civil engineering/engineering geology

rock granular material, grain size distribution, seepage process, mechanical–hydrological–chemical coupling effect, initial compression

**Authors for correspondence:**
Luzhen Wang
e-mail: wangluzhen5@126.com
Hualei Zhang
e-mail: hlzhang1122@126.com

# The variation of grain size distribution in rock granular material in seepage process considering the mechanical–hydrological–chemical coupling effect: an experimental research

Hailing Kong[1,2,3], Luzhen Wang[1] and Hualei Zhang[3]

[1]College of Civil Engineering, and [2]Institute of Coastal Ultra-Soft Soil, Yancheng Institute of Technology, Yancheng, Jiangsu 224051, People's Republic of China
[3]Key Laboratory of Safety and High-efficiency Coal Mining, Ministry of Education, Anhui University of Science and Technology, Huainan, Anhui 232001, People's Republic of China

HK, 0000-0002-0015-9906

As a common solid waste in geotechnical engineering, rock granular material should be properly treated and recycled. Rock granular material often coexists with water when it is used as the filling material in geotechnical engineering. Water flowing in rock granular materials is a complex progress with the mechanical–hydrological–chemical (MHC) coupling effect, i.e. the water scours in the gaps and spaces in the rock granular material structure, produces chemical reactions with rock grains, rock grains squeeze each other under the water pressure and compression leading to re-breakage and producing secondary rock grains, and the fine rock grains are migrated with water and rushed out. In this process, rock grain size distribution (GSD) changes, it affects the physical and mechanical characteristics of the rock granular materials, and even influences the seepage stability of the rock granular materials. To study the variation of GSD in the rock granular material considering the MHC coupling effect after the seepage process, seepage experiments of rock grain samples are carried out and analysed in this paper. The result is expected to have a positive impact on further studies of the properties of the rock granular material.

# 1. Introduction

As a common solid waste in geotechnical engineering, e.g. mining engineering and tunnelling engineering, rock granular material should be properly treated and recycled. Due to the excellent engineering characteristics, such as good compactness, high density, low strength, better seepage ability, low sedimentation, etc., the rock granular, a common construction material, is widely used in a large amount of structures in critical infrastructure systems, such as highways, tunnels, dams and so on [1].

Rock granular material is different from other continuous material; generally, it is composed of rock grains with different sizes, from $10^{-2}$ to $10^2$ mm, different shapes and different arrangements, which results in various porosities, micro-voids and weak boundaries within its structure [2]. Due to the existence of the above-mentioned defects, the mechanical properties of rock granular material will be significantly affected. Relative studies have shown that the mechanical properties of rock granular material are closely related to the mineral composition [3–6], the shape and size of the grains [6–11] and the grain size distribution (GSD) [12–19], etc.

Among the above-mentioned factors, the influences of mineral composition on mechanical properties for rock granular material have been measured through experimental tests and summarized into different regression equations. Li et al. [3] analysed the influence of water and mineral composition on the physical and mechanical properties of phyllite, by making relevant tests on Lu Lin tunnel, and found that the uniaxial compressive strength after water saturation had decreased by about 2.7 times. Zhang and Liu [4] discussed the influence of microstructures and mineral compositions on red-layer mechanical properties; they pointed out that a large amount of fine-grained expansion clay minerals was the main material basis for determining the complex red-clay properties, and it was also an important internal factor which impacted its engineering properties such as water swelling, easily sliming and strong viscosity. Based on the discrete element method (DEM) simulations, Pan et al. [5] found that the piecewise linear relationship between the mineral compositions and the rock mechanical properties was applicable for general rocks.

Either the grain size or the shape of rock granular material had effects on its mechanical properties. Norouzi et al. [10] used a Voronoi element–discrete element method to determine the effects of grains and sampling sizes on their macro/micro-mechanical properties, it showed that the high size-dependency of mechanical properties of intact rocks and the model scale had a more significant impact on macro-mechanical properties than the microstructure pattern. Koyama and Jing [11] used the particle mechanics approaches to study the impacts of sample size and particle size distributions on the mechanical behaviour of rocks, and obtained that the variance of the calculated values of mechanical properties decreased significantly as the side lengths of particle models increase, reaching a representative elementary volume side length ca 5 cm with an acceptable variation of 5%. Hecht [7] considered that highly irregular shaped particles increased the mechanical properties of rocks and building materials, he carried out the shear-box experiments on materials with distinct GSDs and found a remarkable increase of the mechanical strength from non-fractal to fractal mixtures. Jin et al. [8] analysed the effects of the rock block shape on the mechanical properties of cemented soil–rock mixture, and found that cemented soil–rock mixture specimens with 3% cement exhibited evident strain-softening band and localized shear band, whose strength and modulus were greatly increased compared with those of specimen without cement; besides, the peak strength and brittleness index both decreased with the increasing rock block proportion under given conditions. Han et al. [9] proposed a shape factor for granular materials based on particle shape to investigate the influence of particle shape on the mechanical properties of rockfill materials, and found that particle shape greatly affected the particle breakage rate, peak intensity and peak-related internal friction angle for rockfill materials, and the final experimental grading curves all approached the particle breakage grading curve proposed by Einav [20,21].

Scholars recognized that the GSD was particularly important in determining the macroscopic behaviour of granular materials [22–24]. Shimizu et al. [15] investigated the effects of particle number and size distribution on macroscopic mechanical properties of rock models by a series of rock test simulations using a new DEM code, e.g. uniaxial compression test, uniaxial tension test and Brazilian test; it showed that macroscopic mechanical properties of rock model, such as uniaxial compressive strength, Young's modulus and uniaxial tensile strength were significantly affected by porosity of the rock particle number and size distribution, because small particles filled the space among large particles, and the displacement of each particle was restrained by the adjacent particles. Wu et al. [17]

researched the particle size distribution of aggregate effects on the mechanical and structural properties of cemented rockfill; it showed that the cemented rockfill with the overly fine or coarse particle size distribution of aggregate not only presented the more deteriorative microstructures but also had the more early damaged areas with active AE signals and more initiated cracks during loading. Mahdevari & Maarefvand [19] studied the effects of volumetric block proportion, maximum block size and distribution function on the compressive strength and failure patterns of these rocks by executing the unconfined compressive test on large-scale synthetic samples, and they found that the compressive strength of block in matrix rocks was found to have a direct relation with volumetric block proportion and maximum block size and an indirect one with fractal dimension, the fracture patterns of remoulded samples were related to the volumetric block proportion and the fractal dimension. Besides, the GSD also played an important role in other properties of the rock granular material, such as deformation [25–27] and permeability [28–31].

In geotechnical engineering, water often coexists with the rock granular materials. Water flows in the gaps and spaces of the rock granular materials [32], scours and erodes them, makes some of the rock granular materials re-break and produces secondary rock grains [33], drives particles to migrate and loss through the gaps and spaces [34–39], which varies the porosity [40,41], local stress [41] and seepage fields [40,42–49]. As the rock grains' migration and loss continues, porosity increases and permeability grows until mutation, resulting in seepage instability and even seepage catastrophe. Seepage in rock granular material is a complex process with the mechanical–hydrological–chemical (MHC) coupling effect [50,51]. Due to the MHC coupling effect on the rock granular material, the GSD of the rock granular material varies over time in the seepage process. The arrangement of the rock grains redistributes, associated with the macroscopic behaviours in the geotechnical engineering, the redistribution may affect rock grains' strength, porosity, permeability and excess pore pressure, etc.; it also may increase the deformation (or rate of deformation) of the engineering structure to a greater degree than is predicted, and cause structural instability [52]. Therefore, the variation of GSD in the rock granular material due to the coupling effect of re-breakage induced by mechanical deformation (M), mass loss induced by hydraulic flow (H) and water–rock chemical reaction (C) in seepage process deserves attention.

Consequently, seepage experiments of rock granular materials combining the MHC coupling effect are carried out in this paper. The mass of rock grains of different sizes after experiments under different compressions is measured and compared with the original ones. The stability is discussed from the variation of the GSD in the process. The ratio $d_i/d_M$ is introduced to express the varied GSD after seepage experiments. Referring to the Talbot continuous grading formula, the expression of the varied GSD is fitted, derived and verified. The results will be of great significance to promoting research on the application of the rock granular materials in geotechnical engineering.

# 2. Experimental materials and methods

## 2.1. Experimental system

The used seepage experimental system for the rock granular material which can realize mass loss is designed and established by ourselves (figure 1) [53], which includes four parts, an axial-loading and displacement-controlling subsystem, a fluid flow subsystem, a fine rock grains collection subsystem and a data acquisition-analysis subsystem.

The axial-loading and displacement-controlling subsystem contains a 30 ton hydraulic material testing machine, a variable displacement piston pump, a reversing valve and a relief valve to control the power, a single-action hydraulic cylinder, a displacement transducer to control the displacement of the samples.

The fluid flow subsystem is the core part of the experimental system, it contains a permeameter that is filled with the sample, a variable displacement piston pump and a quantitative displacement piston pump to provide power to drive the water flow, reversing valves and relief values to change the power, a double-action hydraulic cylinder to realize a reality water seepage test through driving water flow by oil pressure, a pressure transducer and a flow transducer to gather data [54]. The permeameter is designed according to the ASTM Standard Practice for Making and Curing Concrete Test Specimens in the Laboratory [55], and it is the most important component of the fluid flow subsystem, which contains several parts, such as a floor, cylinder, permeable plate, piston, overflow tank and the lost fine rock grains collecting tank, as shown in figure 2. The inner diameter of the

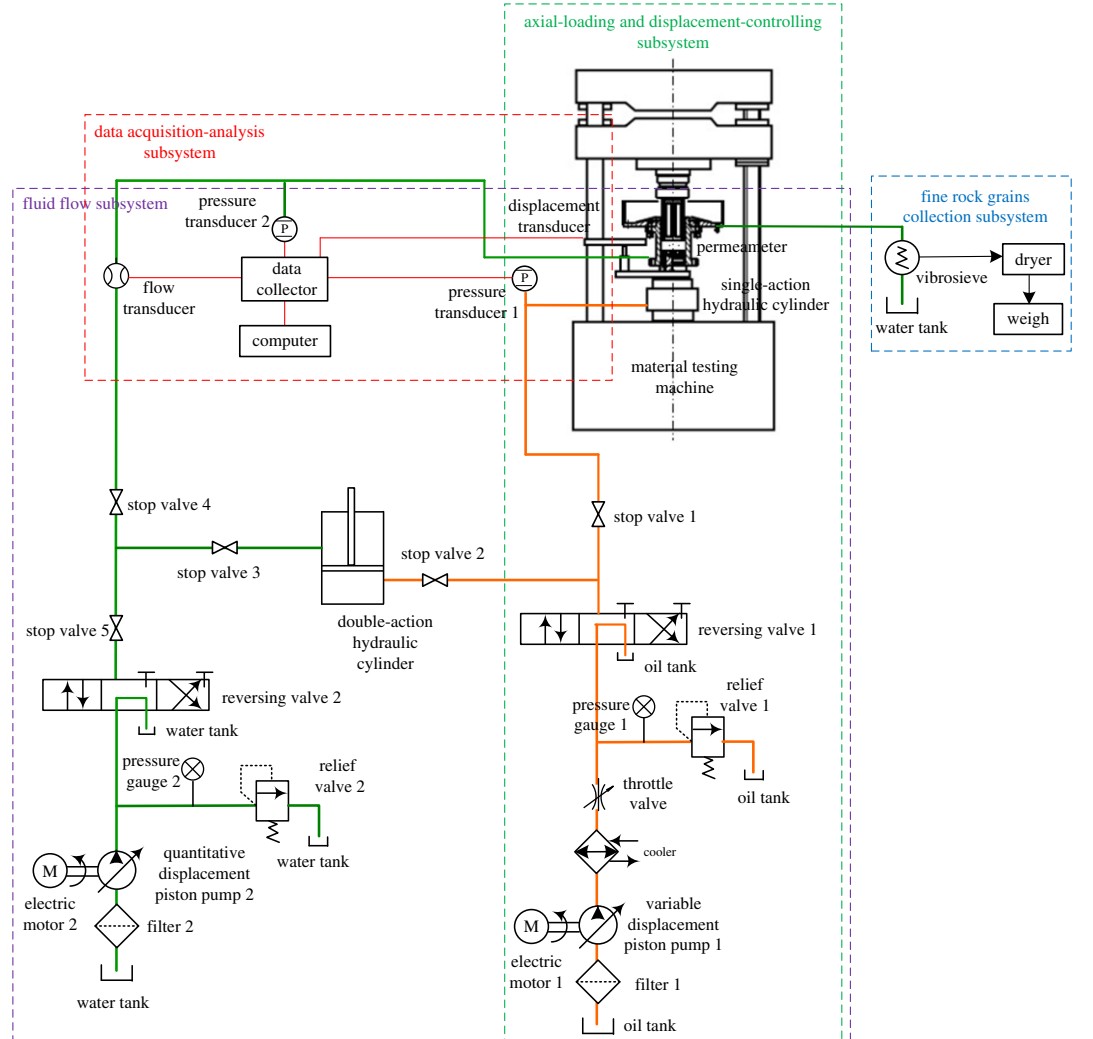

**Figure 1.** Experimental system.

cylinder is 100 mm, and its height is 200 mm; the height–diameter ratio is 2 : 1, which coincides with the ASTM standard [56]. The piston with a height of 25 mm has 19 holes with a diameter of 10 mm on it, looking like a honeycomb briquette. The permeable plate is set at the bottom of the sample in the cylinder to keep the water flow stable, they also have many holes, whose diameter is 2 mm. Six large exits with a height of 115 mm are around the cylinder of the overflow tank. The unique design of the piston and overflow tank makes sure the fine rock grains can rush out freely, which is the key component of the permeameter to realize permeate in the rock granular material accompanying mass loss. The lost fine rock grains collecting tank has an outlet to make sure the crushed out fine rock grains flow with water through the pipe to the fine rock grains collection subsystem.

The fine rock grains collection subsystem contains a vibrosieve, a filter screen of 300 meshes, a dryer, etc. Fine rock grains are collected by the vibrosieve and filter screen when the fine rock grains migrate with water flow and rush out of the permeameter, then they are dried and weighed.

The data acquisition-analysis subsystem consists of a data collector, a computer and transducers. When testing, the physical parameters (e.g. pressure and water flow) are acquired by transducers, collected by the data collector and recorded by the computer.

## 2.2. Experimental materials and samples

The rock granular material used in our seepage experiments is screened from mudstone after crushing, which is collected from Changcun Coal Mine, Lu'an Mining Bureau, Shanxi Province, China. Its mineral compositions are usually quartz ($SiO_2$), illite ($KAl_2(OH)_2(AlSi)_4O_{10}$), kaolinite ($Al_4(OH)_8Si_4O_{10}$),

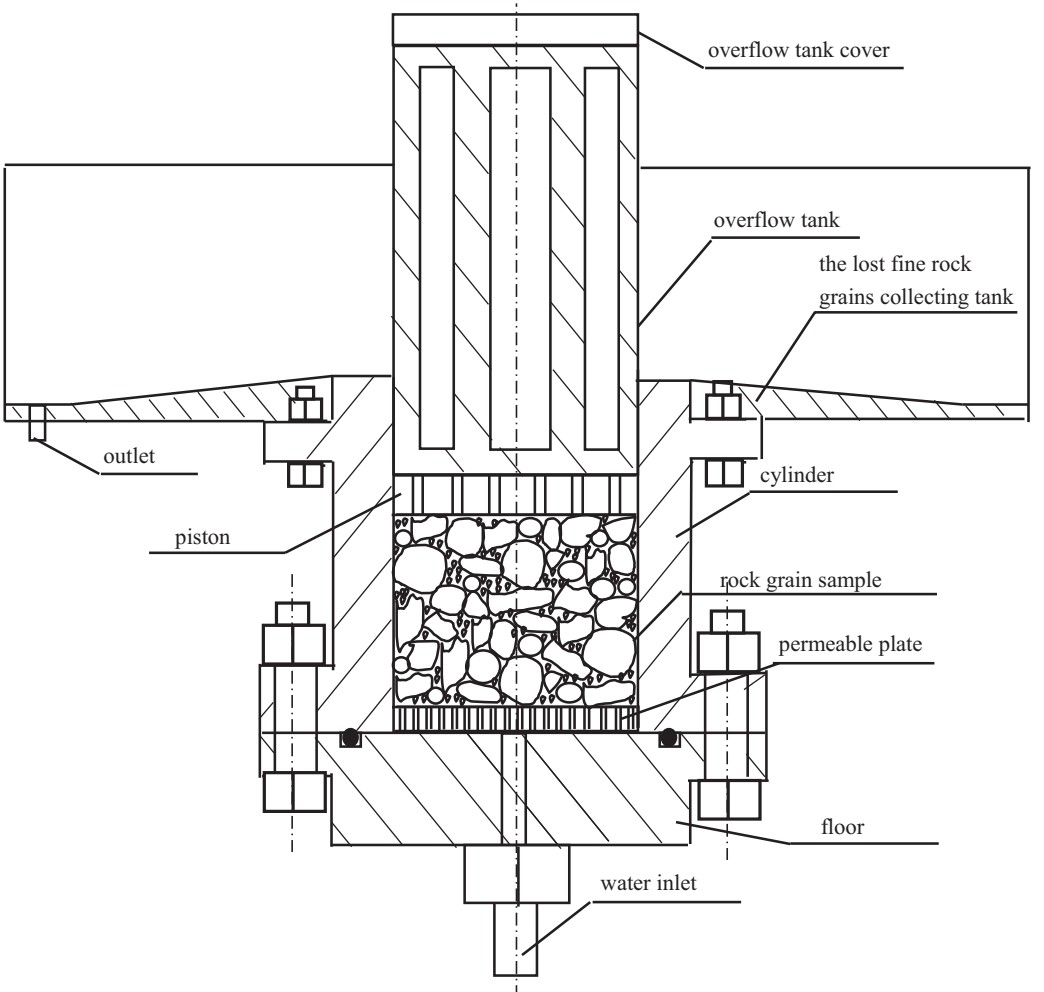

**Figure 2.** Permeameter for the rock grains realizing the mass loss.

muscovite (KAl$_3$Si$_3$O$_{10}$(OH)$_2$), calcite (CaCO$_3$), dolomite (CaMg(CO$_3$)$_2$), gypsum (CaSO$_4$ or CaSO$_4$·2H$_2$O), etc. [57–59].

ASTM suggests that the minimum diameter of a cylindrical sample must be at least three times the maximum rock grain size for eliminating the size effect [55]. In our experiments, the ratio is set as four times, which satisfies the ASTM standard, so that the maximum size of the rock grains is 25 mm. Samples in seepage experiments are always composed of rock grains of several sizes in a fixed proportion [60–63], such as 1 : 1 : 1 : 1, etc. However, in our rock grain seepage experiments, continuous gradation is adopted, i.e. a set of standard sieves with specified sieve size are used to screen the mixed rock grains. The size curve of the mixed rock grains is smooth and continuous, and there is a certain proportion relationship between the adjacent rock grain sizes. Therefore, the well-known Talbot continuous grading formula [64] is adopted, and the rock grains are screened into eight groups, 0–2.5 mm, 2.5–5 mm, 5–8 mm, 8–10 mm, 10–12 mm, 12–15 mm, 15–20 mm and 20–25 mm before the experiments, as seen in figure 3.

The Talbot continuous grading formula is

$$p_0(d \leq d_i) = \left(\frac{d_i}{d_M}\right)^n \times 100\%, \tag{2.1}$$

where $p_0(d \leq d_i)$ is the original mass ratio of the rock grains with the size not larger than $d_i$, $d_i$ is the largest grain size in the group $i$, $d_M$ is the largest grain size in the sample and $n$ is the Talbot power exponent (TPE), which is set as 0.1, 0.2, 0.3, … ,1.0 in our experiments, and the percentage curves of the rock grains in samples with different TPEs are shown in figure 4.

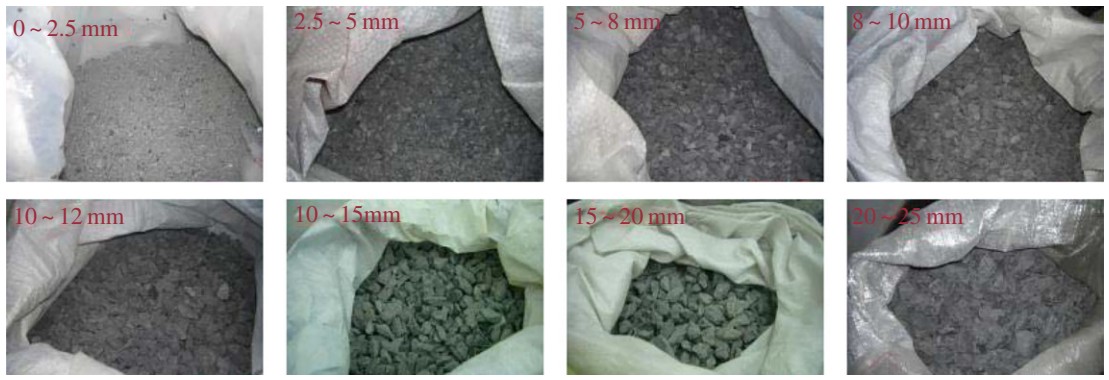

**Figure 3.** Groups of the rock grains.

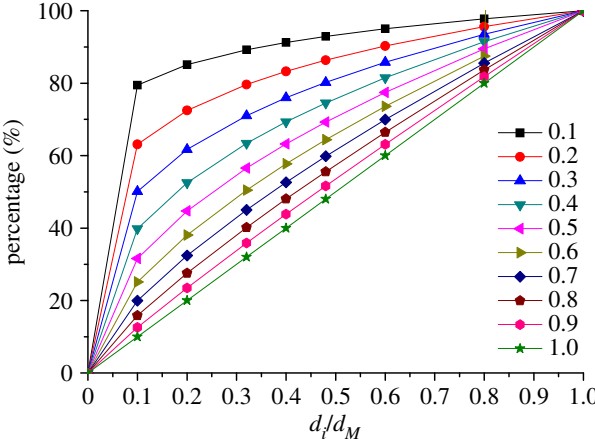

**Figure 4.** Percentage curves of the test samples.

The mass of rock grains of different sizes could be calculated as

$$M(d_i \leq d \leq d_{i+1}) = \left[ \left( \frac{d_{i+1}}{d_M} \right)^n - \left( \frac{d_i}{d_M} \right)^n \right] M_T, \tag{2.2}$$

where $M(d_i \leq d \leq d_{i+1})$ is the mass of rock grains with the size between $d_i$ and $d_{i+1}$, and $M_T$ is the total mass of the samples, based on the size of the cylinder, which is 2000 g each. The mass of rock grains of different sizes is listed in table 1.

## 2.3. Experimental flow

Before the experiment, experimental system installation and debugging are going to ensure that the pore pressure can research the default, no leakage exists in the permeation circuit, and the data collector and transducers work normally, etc.

Samples are prepared before testing. We mix kinds of rock grains with different sizes after weighing based on table 1, then charge the mixture into the permeameter. The loading rate in our experiment was 1 mm min$^{-1}$. In order to analyse the influence on the variation of GSD of initial compression in the rock grains' seepage process, the charged samples are compressed to a certain degree by axial loading. In our tests, the axial-loading displacement is 0, 10, 20, 30 and 40 mm. Because the mass of rock grains will have obvious variation when the loading gradient is set every 10 mm through the pre-tests, the loading gradient of 5 mm could not achieve such a significant effect, and the discrimination is not significant enough. If the loading gradient is set as every 15 or 20 mm, it is difficult to achieve three to four different compaction quantities in the test, and it is difficult to compare, too. So, the loading gradient of 10 mm is relatively reasonable. Some other samples are charged without initial compression. In the following sections, OS means the original sample, WCS means sample without initial compression, C10S, C20S, C30S and C40S, respectively, means a sample with initial compression of 10, 20, 30 and 40 mm.

**Table 1.** Mass of rock grains under different TPEs (g).

| rock grain size (mm) | TPE | | | | | | | | | |
|---|---|---|---|---|---|---|---|---|---|---|
| | 0.1 | 0.2 | 0.3 | 0.4 | 0.5 | 0.6 | 0.7 | 0.8 | 0.9 | 1.0 |
| 0–2.5 | 1588.7 | 1261.9 | 1002.4 | 796.2 | 632.5 | 502.4 | 399.1 | 317.0 | 251.8 | 200.0 |
| 2.5–5 | 114.0 | 187.6 | 231.7 | 254.4 | 262.0 | 259.1 | 249.2 | 234.9 | 218.1 | 200.0 |
| 5–8 | 81.9 | 142.9 | 186.9 | 217.3 | 236.9 | 248.1 | 252.6 | 251.9 | 247.4 | 240.0 |
| 8–10 | 40.3 | 72.7 | 98.4 | 118.4 | 133.5 | 144.6 | 152.3 | 157.1 | 159.5 | 160.0 |
| 10–12 | 33.6 | 61.8 | 85.4 | 104.9 | 120.7 | 133.4 | 143.4 | 150.9 | 156.3 | 160.0 |
| 12–15 | 41.9 | 78.8 | 111.1 | 139.2 | 163.6 | 184.5 | 202.3 | 217.3 | 229.8 | 240.0 |
| 15–20 | 55.5 | 106.9 | 154.7 | 198.8 | 239.7 | 277.3 | 312.0 | 343.9 | 373.2 | 400.0 |
| 20–25 | 44.1 | 87.3 | 129.5 | 170.8 | 211.1 | 250.6 | 289.2 | 327.0 | 363.9 | 400.0 |

As the permeameter is assembled into the experimental system, in order to simulate the state of crushed rock mass in engineering, the samples would be pre-loaded at a certain pressure. On the one hand, this pressure cannot make the structure of crushed rock mass deform, on the other hand, it cannot make the crushed rock particles compress and destroy. Through many experiments, the pressure applied can meet both of the above requirements when it does not exceed 0.02 MPa. After loading initial pressure, the set displacement then is loaded. Subsequently, water is injected into the permeameter for half an hour until the sample is saturated.

When the seepage test starts, we open the quantitative displacement piston pump and inject water into the double-action hydraulic cylinder. Then, the piston of the double-action hydraulic cylinder is driven by oil, and water is injected into the sample again. After adjusting the water pressure to the set value, water starts to permeate through the sample for 18 000 s to record this seepage process as completely as possible, water pressure and flow are acquired and recorded in real time [53].

The lost fine rock grains are collected at regular intervals, then the collected fine rock grains are numbered, dried, screened and weighed. At the end of the tests, the residual sample is discharged, then dried, screened and weighed as well.

In order to ensure the reliability and accuracy of the experimental data, three items are taken for each seepage experiment, and their average values are used as the final experimental data.

# 3. Experimental results and analysis

During the seepage process, water and rock interact with each other, rock grains are squeezing and rubbing each other, larger rock grains are re-broken and new secondary grains are produced, the fine rock grains migrate with water flow, some of them rush out of the cylinder, therefore the number of grains of each size changes, i.e. the GSD changes.

## 3.1. Mass variation before and after seepage experiments

In order to analyse the effect on the mass variation of initial compression, the masses were compared between the original samples and the residual samples, including samples without initial compression and with initial compression. Due to the limitation of the length of the article, it was impossible to give a detailed description of the variation under various compressions, so only samples with initial compression of 10 mm were selected to describe in this manuscript.

### 3.1.1. The residual masses in samples with different initial compressions

A new physical quantity, the residual mass ratio is introduced to compare the mass distribution before and after seepage experiments, which is defined as

$$r_r(d_i) = \frac{m_r(d_i \leq d \leq d_{i+1})}{m_o(d_i \leq d \leq d_{i+1})} \times 100\%, \tag{3.1}$$

where $m_r(d_i \leq d \leq d_{i+1})$ is the mass of rock grains whose size is between $d_i$ and $d_{i+1}$ in the residual samples after tests, and $m_o(d_i \leq d \leq d_{i+1})$ is the mass of rock grains whose size is between $d_i$ and $d_{i+1}$ in the original samples before experiments.

The residual mass ratios of samples without initial compression are presented as the black columns, while the ones with initial compression are presented in red, which are shown in figure 5.

### 3.1.2. The mass variation of rock grains in samples without initial compression

For samples without initial compressions, the rock grains with the size of 20–25 mm reduce after experiments, and basically, samples with larger TPEs have a larger residual mass ratio, e.g. the residual mass ratios of samples with larger TPEs, from 0.6 to 1.0 (except for 0.7) are more than 75%, especially samples with TPEs of 0.9 and 1.0, their residual mass ratios reach 99% and 95%, respectively, while the residual mass ratio of samples with TPE of 0.7 only 21%. The rock grains with the size of 15–20 mm become more numerous after experiments, except for samples with TPEs of 0.3, 0.5 and 0.9, whose residual mass ratios are 78%, 46% and 71%, and the maximum residual mass ratio reaches 162% in samples with TPE of 0.1. The rock grains with the size of 12–15 mm become less after experiments, except for samples with TPEs of 0.3 and 0.5, and their residual mass ratios are between 48% and 90%.

If combining the curves of the residual mass ratio of samples with the size of 12–15, 15–20 and 20–25 mm under different TPEs together, a very interesting phenomenon will be found. The curves are divided into two parts, Part 1 and Part 2, as shown in figure 6, they are separated by a line between the TPEs of 0.6 and 0.7. In Part 1, the curves of residual mass ratio for grains of 20–25 and 15–20 mm have the same trends of change except for an irregular point, which is marked in the figure, and both of them have the opposite tendency with that of 12–15 mm. In Part 2, the curves of the residual mass ratio for grains of 12–15 and 15–20 mm have the same trends of change, and they have the opposite tendency with the curve for rock grains of 20–25 mm.

After experiments, the number of rock grains with the size of 10–12 mm decreases except for samples with TPEs of 0.5, 0.7 and 0.8, whose residual mass ratios are 120%, 126% and 103%. The amount of the rock grains with the size of 8–10 mm changes little, their residual mass ratios are from 89% to 111%, except for samples with TPE of 0.5, whose residual mass ratio reaches 165%. The amount of rock grains with the size of 5–8 mm also changes little, their residual mass ratios are between 92% and 107%.

When putting the curves of residual mass ratio of samples with the size of 5–8, 8–10 and 10–12 mm under different TPEs together, as shown in figure 7, the curves of residual mass ratio for grains of 5–8 and 8–10 mm have the same trends of change, but the curve of 10–12 mm has three different variations with the other two curves, which are marked in orange. Actually, the residual mass ratio of the rock grains with the size of 10–12 mm increases with TPE increasing when TPE is less than 0.6 (except for 0.2, which is marked as an irregular point in the figure), but it decreases with TPE increasing when TPE is larger than 0.6, the two trend lines are illustrated as pink dashed lines in figure 7.

The number of rock grains with the size of 2.5–5 mm increases, their residual mass ratios are from 104% to 124%, except for samples with TPEs of 0.1, 0.3 and 0.4, whose residual mass ratios are 57%, 97% and 98%. The number of rock grains with the size of 0–2.5 mm decreases, their residual mass ratios are from 44% to 93%, except for samples with TPE of 0.9, whose residual mass ratio is 103%.

When putting the curves of the residual mass ratio of samples with the size of 0–2.5 and 2.5–5 mm under different TPEs together, as shown in figure 8, in general tendency, the residual mass ratios of rock grains with the size of 0–2.5 and 2.5–5 mm both increase with TPE increasing except for samples with TPE of 1.0. The curves are divided into three parts, in Part 1, both of the two curves have a huge growth, 36% for rock grains of 0–2.5 mm and 47% for the other; in Part 2, they fluctuate and then increase, one for 17% and the other for 20%; in Part 3, both of them fall back about 12%–13%.

### 3.1.3. The mass variation of rock grains in samples with initial compression of 10 mm

For samples with initial compression of 10 mm, rock grains with the size of 20–25 mm reduce after the experiments (even to zero, e.g. samples with TPEs of 0.2 and 0.5), their residual mass ratios are between 8% and 56% (more accurately, they are between 8% and 37%, only samples with TPE of 1.0 has the ratio of 56%); in general tendency, the residual mass ratio is larger when the samples' TPE is larger, except for samples with TPEs of 0.1, 0.5 and 0.7. The rock grains with the size of 15–20 mm become more numerous, and their residual mass ratios are from 106% to 145%. The residual rock grains with the

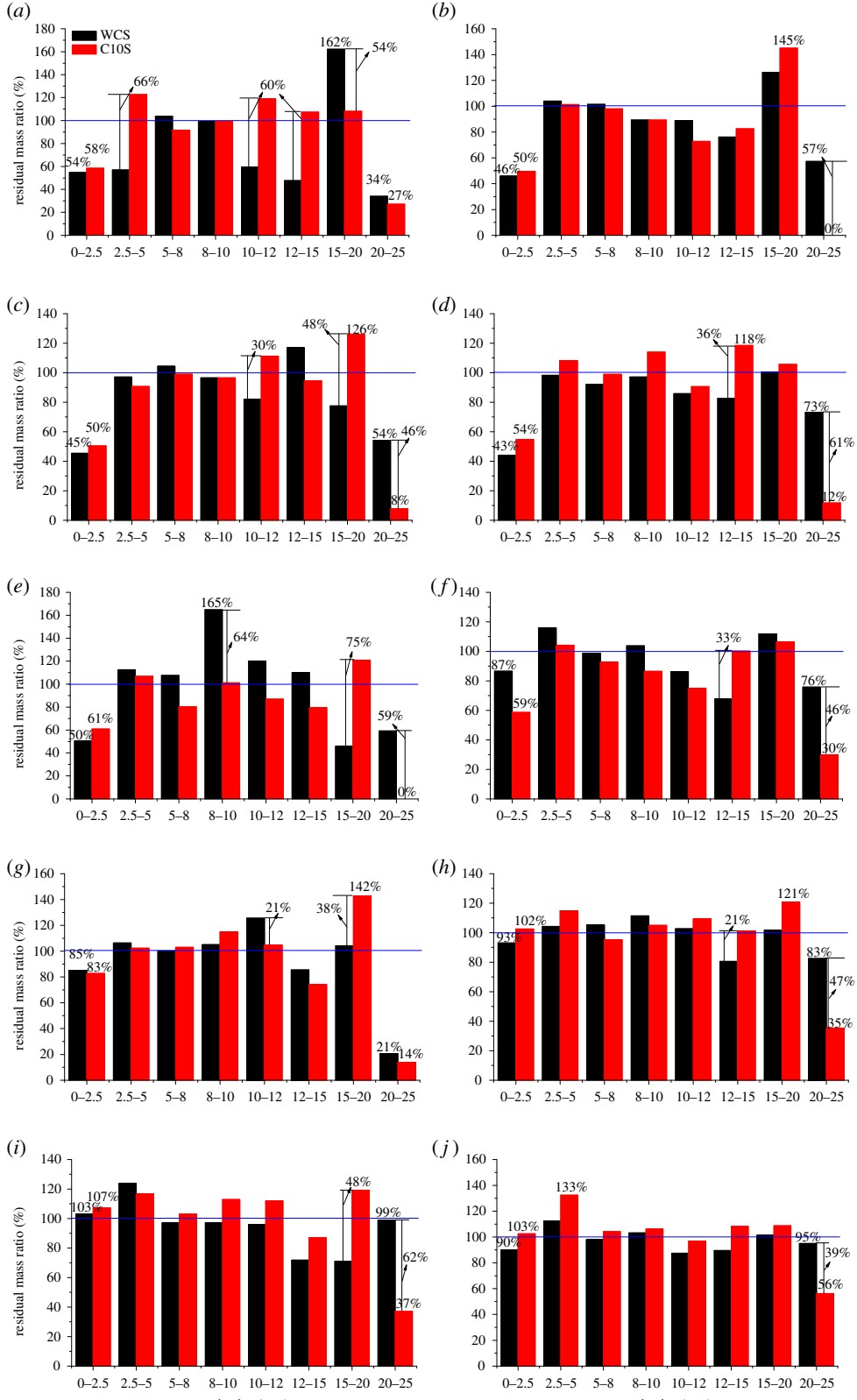

**Figure 5.** The residual mass ratio. TPE: (*a*) 0.1, (*b*) 0.2, (*c*) 0.3, (*d*) 0.4, (*e*) 0.5, (*f*) 0.6, (*g*) 0.7, (*h*) 0.8, (*i*) 0.9 and (*j*) 1.0.

size of 12–15 mm increase or decrease during the experiments, their residual mass ratios are between 74% and 119%.

When we draw the curves of residual mass ratio for rock grains of 12–15, 15–20 and 20–25 mm in samples with different TPEs together, as presented in figure 9, it is obvious that the curves of 12–15

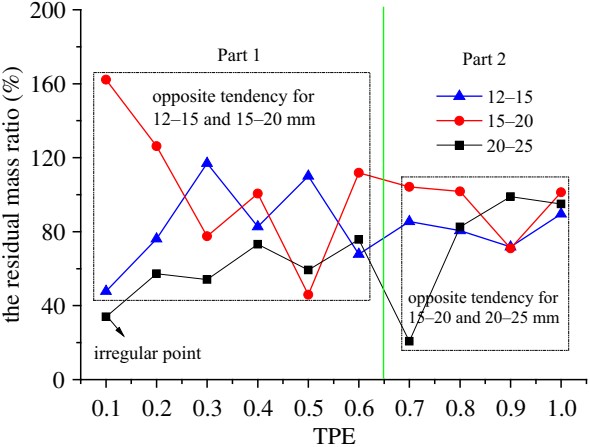

**Figure 6.** The curves of the residual mass ratio for rock grains of 12–15, 15–20 and 20–25 mm.

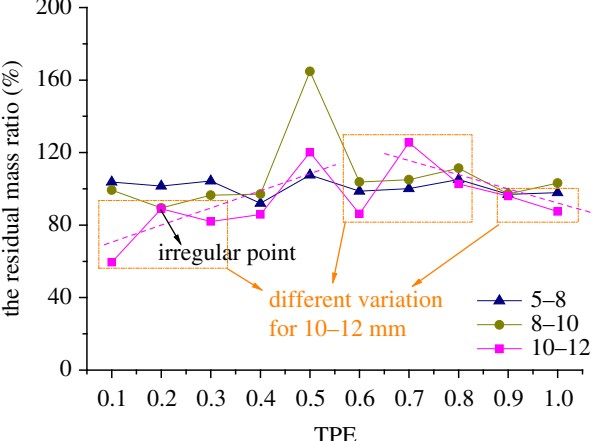

**Figure 7.** The curves of the residual mass ratio for rock grains of 5–8, 8–10 and 10–12 mm.

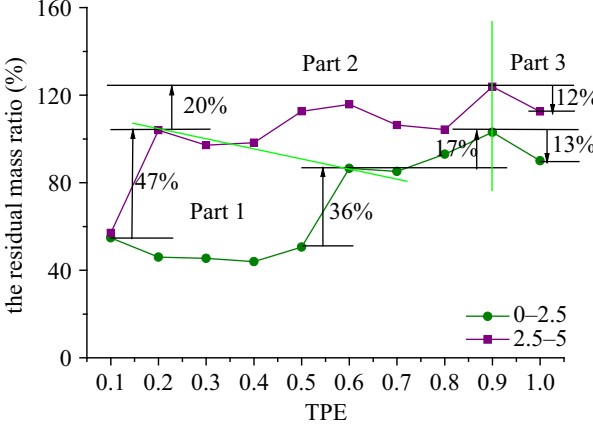

**Figure 8.** The curves of the residual mass ratio for rock grains of 0–2.5 and 2.5–5 mm.

and 20–25 mm have the same change tendency, while they have the opposite change tendency with the curve of 15–20 mm. Especially, in samples with TPEs from 0.7 to 1.0, the residual mass ratios of 15–20 and 20–25 mm have a completely opposite trend, the former decreases with the increase of TPE, while the latter increases, which are illustrated as dashed lines in figure 9.

The rock grains with the size of 10–12 mm change less than 20% during the experiments, it increases in samples with TPEs of 0.1, 0.3, 0.7, 0.8 and 0.9, while decreasing in other samples. The residual rock

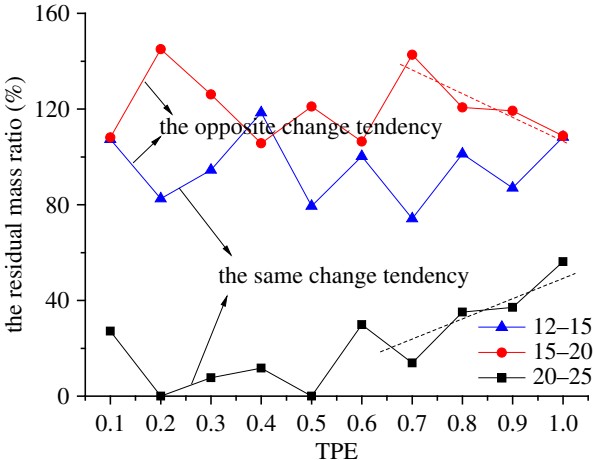

**Figure 9.** The curves of the residual mass ratio for rock grains of 12–15, 15–20 and 20–25 mm.

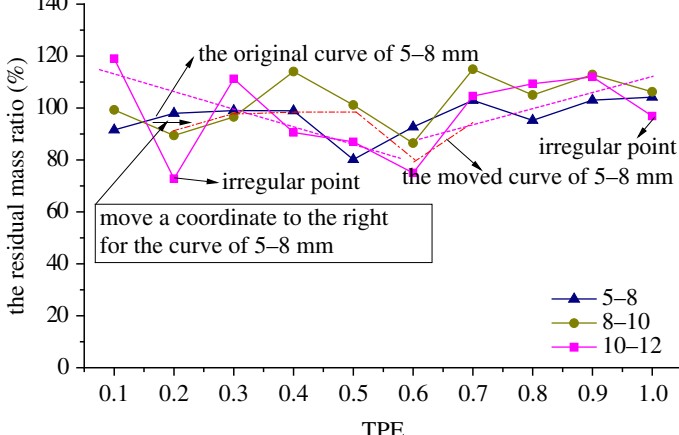

**Figure 10.** The curves of the residual mass ratio for rock grains of 5–8, 8–10 and 10–12 mm.

grain with the size of 8–10 mm decreases in samples with TPEs from 0.1 to 0.3, while it increases from 0.4 to 1.0 except for 0.6, and their residual mass ratios are between 86% and 115%. The amount of rock grains with the size of 5–8 mm changes little during the experiments, their residual mass ratios are from 92% to 104%, except for samples with TPE of 0.5, whose residual mass ratio is smaller only 80%; and the rock grain decreases in samples with TPEs from 0.1 to 0.6, and increases from 0.7 to 1.0 except for 0.8.

Figure 10 combines the curves of residual mass ratio for rock grains of 5–8, 8–10 and 10–12 mm in samples with different TPEs together; it shows that the curves of 10–12 and 8–10 mm have the same change tendency, and the curve of 5–8 mm would have the same tendency with the other two if we move a coordinate to the right direction for one part of the curve of 5–8 mm; the corresponding abscissa ranges from 0.1 to 0.6, the original curve and the moved one illustrated as a red dashed line are noted in figure 10. Meanwhile, the residual mass ratio of the rock grains with the size of 10–12 mm decreases with TPE increasing when TPE is less than 0.6 except for 0.2 (which is marked as an irregular point in the figure), while it increases with TPE increasing when TPE is larger than 0.6 except for 1.0 (marked as the irregular point as well), we can also find tendencies as pink dashed lines in figure 10.

The number of rock grains with the size of 2.5–5 mm increases during the experiments, their residual mass ratios are from 101% to 133%, except for samples with TPE of 0.3, whose residual mass ratio is 91%. The number of rock grains with the size of 0–2.5 mm decreases in samples with TPEs of 0.1–0.7, whose residual mass ratios are from 50% to 83%, while it increases in samples with TPEs of 0.8–1.0, whose residual mass ratios are between 103% and 107%.

If we put the curves of the residual mass ratio of samples with the size of 0–2.5 and 2.5–5 mm under different TPEs together in figure 11, it could be found that, in general tendency, with TPE increasing, the

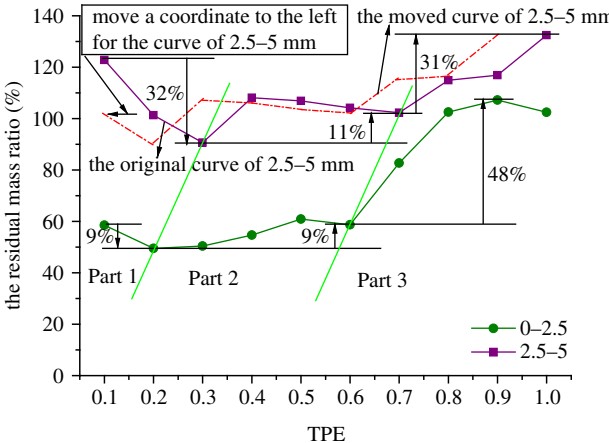

**Figure 11.** The curves of the residual mass ratio for rock grains of 0–2.5 mm and 2.5–5 mm.

**Table 2.** The statistics of the changes of rock grain size when the initial compression is 10 mm.

| rock grain size (mm) | TPE | | | | | | | | | |
|---|---|---|---|---|---|---|---|---|---|---|
| | 0.1 | 0.2 | 0.3 | 0.4 | 0.5 | 0.6 | 0.7 | 0.8 | 0.9 | 1.0 |
| 0–2.5 | L | L | L | L | L | S | S | L | L | L |
| 2.5–5 | L | S | S | L | S | S | S | L | S* | L |
| 5–8 | S | S | S | L | S | S | L | S* | L | L |
| 8–10 | U | U* | U* | L | S | S | L | S* | L | L |
| 10–12 | L | S | L* | L | S | S | S | L | L | L |
| 12–15 | L | L | S | L | S | L | S | L | L | L |
| 15–20 | S | L | L | L | L | S | L | L | L | L |
| 20–25 | S | S | S | S | S | S | S | S | S | S |

residual mass ratios of rock grains with the size of 0–2.5 and 2.5–5 mm both decrease and then increase. Figure 11 is also divided into three parts, in Part 1, both of the two curves decrease, 9% for rock grains of 0–2.5 mm 32% for the other; in Part 2, they fluctuate and then increase, one for 9% and the other for 11%; in Part 3, both of them consistently grow for 48% and 31%, respectively. We can also discover in figure 11 that these two curves have a similar change tendency, and they would have the completely same tendency if we move a coordinate to the left direction for the curve of 2.5–5 mm.

## 3.2. The variation of GSD in samples with different initial compressions

Scientists found that compression and grain size have interaction with each other [65]. From the above sections, it could be found that initial compression has its own effect on the experimental results, which is discussed in this section by two steps, in the first step, we will compare the experimental results between samples without initial compression and with initial compression of 10 mm, which are analysed separately in the preceding paragraphs; in the second step, we will discuss the effect of the amount of the initial compression on the experimental results.

As seen in figure 5, the influences on the residual mass ratio of initial compression are three kinds, either becoming larger, smaller or unchanging. We use 'L' to stand for larger, 'S' for smaller and 'U' for unchanging to gather statistics of the changes in figure 5, and the results are listed in table 2.

It could be found in table 2 that, comparing with the samples without initial compression, these ones with initial compression of 10 mm have significant changes in the experimental results, for example, the residual mass ratio of rock grains with size of 0–2.5 mm increases except for samples with TPEs of 0.6 and 0.7, as well as the rock grains of 10–12 mm except for samples with TPEs of 0.2, 0.5, 0.6 and 0.7, the rock

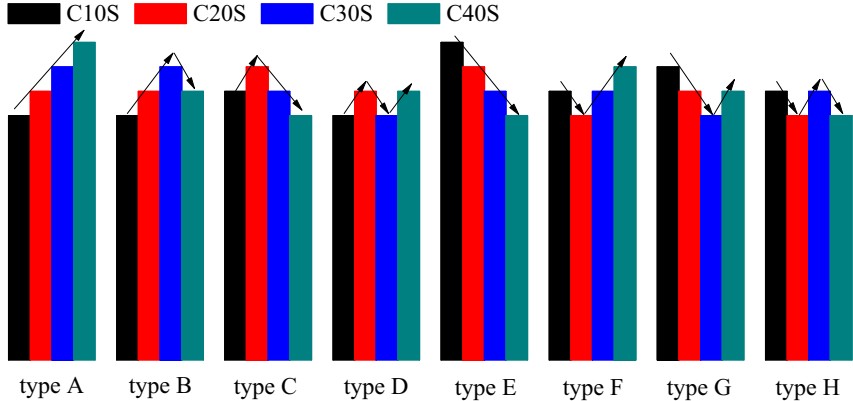

**Figure 12.** The type of the influence of initial compression.

**Table 3.** The statistics of types.

| rock grain size (mm) | TPE | | | | | | | | | |
|---|---|---|---|---|---|---|---|---|---|---|
| | 0.1 | 0.2 | 0.3 | 0.4 | 0.5 | 0.6 | 0.7 | 0.8 | 0.9 | 1.0 |
| 0–2.5 | H | H | H | F | G | C | B | A | D | A |
| 2.5–5 | B | H | D | H | F | A | F | H | D | E |
| 5–8 | B | C | E | E | B | B | C | C | A | H |
| 8–10 | B | B | G | H | G | D | H | E | G | G |
| 10–12 | H | C | E | E | G | D | E | E | H | B |
| 12–15 | H | B | E | H | C | E | B | H | F | G |
| 15–20 | G | H | E | D | E | G | E | E | E | G |
| 20–25 | D | C | F | C | A | C | H | C | C | B |

grains of 12–15 mm except for samples with TPEs of 0.3, 0.5 and 0.7, and the rock grains of 15–20 mm except for samples with TPEs of 0.1 and 0.6, which results from the promotion of initial compaction to re-breakage behaviour, which produces more secondary rock grains, and it also causes the residual mass ratio of rock grains of 20–25 mm to be smaller than that in samples without initial compression. The maximum increase of the residual mass ratio induced by the initial compression reaches 75% in samples with TPE of 0.5.

It also could be found that the residual mass ratios of rock grains (except for grains of 20–25 mm) are larger in larger TPEs samples with initial compression of 10 mm, such as 0.8, 0.9 and 1.0, as seen in the gold part in table 2; the ratios of rock grains from 2.5 to 12 mm are smaller in samples with TPEs from 0.2 to 0.6 (except for 0.4) than that in samples without initial compression; see the blue parts.

The initial compression in samples has indeed influence on the mass variation of rock grains, so, does the amount of the initial compression have a similar influence on the experimental results? It will be discussed in the following content.

The influences of the initial compressions on the experimental results have eight different types, as shown in figure 12, we define them as types A to H. We observe the changes induced by the initial compression in samples after the experiments, gather the statistics of these types in table 3, in which the types are simplified as 'A' to 'H'.

It could be discovered from table 3 that rock grains of each size have different variation in the experimental process, in which the variation of H accounts for one-fifth, E accounts for 15 times, B and C account for 11 times, G accounts for one-eighth, D accounts for 7 times, A and F account for one-sixteenth each.

The variation of H occurs in samples with TPEs except 0.5 and 0.6, especially concentrates in samples with TPEs of 0.1, 0.2 and 0.4. H is observed in each rock grain size and its probability reaches 30% in rock grains with the size of 0–5 mm and 20% in 8–15 mm. E occurs in samples with TPEs except 0.1 and 0.2, especially concentrates in samples with TPEs of 0.3 and 0.8, E is observed in rock grains with the size of

**Table 4.** The mass variation of rock grains with a size of 0–2.5 mm (g).

| samples | TPE 0.1 | 0.2 | 0.3 | 0.4 | 0.5 | 0.6 | 0.7 | 0.8 | 0.9 | 1.0 |
|---|---|---|---|---|---|---|---|---|---|---|
| 0S | 1588.7 | 1261.9 | 1002.4 | 796.2 | 632.5 | 502.4 | 399.1 | 317.0 | 251.8 | 200.0 |
| WCS | 870.0 | 580.0 | 455.0 | 350.0 | 320.0 | 435.0 | 340.0 | 295.0 | *260.0* | 180.0 |
| C10S | 930.0 | 625.0 | 505.0 | 435.0 | 385.0 | **295.0** | **330.0** | 325.0 | 270.0 | 205.0 |
| C20S | 845.0 | 520.0 | 400.0 | 410.0 | 385.0 | 415.0 | 380.0 | 340.0 | 280.0 | 223.0 |
| C30S | 1275.0 | 565.0 | 405.0 | 430.0 | 370.0 | 375.0 | 495.0 | 390.0 | 250.0 | 270.0 |
| C40S | 1165.0 | 385.0 | 400.0 | 490.0 | 410.0 | 300.0 | 390.0 | 405.0 | 395.0 | 305.0 |

2.5–20 mm, and its probability reaches 50% in rock grains with the size of 15–20 mm and 40% in 10–12 mm. B occurs in samples with TPEs except 0.3, 0.4, 0.8 and 0.9, especially concentrates in samples with TPE of 0.1, it is mainly observed in rock grains with the size of 0–15 mm, and its probability reaches 30% in rock grains with the size of 5–8 mm. C occurs in samples with TPEs except 0.1, 0.3 and 1.0, especially concentrates in samples with TPE of 0.2, it is mainly observed in rock grains with the size of 5–8, 10–15 and 20–25 mm, and its probability reaches 50% in rock grains with the size of 20–25 mm. G occurs in samples with TPEs of 0.1, 0.3, 0.5, 0.6, 0.9 and 1.0, especially concentrates in samples with TPEs of 0.5 and 1.0, it is mainly observed in rock grains with the size of 8–20 mm, and its probability reaches 40% in rock grains with the size of 8–10 mm. D occurs in samples with TPEs of 0.1, 0.3, 0.4, 0.6 and 0.9, and it is mainly observed in rock grains with the size of 0–5, 8–12 and 15–25 mm. A and F mainly occur in smaller rock grains in samples with TPEs of 0.6–1.0 and 0.4–0.7, respectively.

Because of the complexity of rock grain arrangement in the cylinder and the relatively small number of items for samples with different TPEs under different initial compressions, no other obvious regularity can be found through this statistical distribution in table 3.

## 3.3. The MHC coupling effect on the mass variation of rock grains

In this section, we want to analyse the MHC coupling effect on the mass variation of rock grains in the experimental process, including mass loss, re-breakage and water–rock chemical reactions.

### 3.3.1. The effect of mass loss in the seepage process

If we want to study the effect of mass loss, it is necessary to analyse the residual masses of rock grains with the size of 0–2.5 mm, which are listed in table 4, because the decrease of the mass of rock grains with the size of 0–2.5 mm completely results from mass loss.

It could be found in table 4 that the amount of the lost mass in the experimental process is very large. The mass loss is the key factor of the mass variation and the GSD variation in the samples without initial compression in the seepage process.

Comparing the masses in WCS and C10S, the residual masses in most samples increase except for samples with TPEs of 0.6 and 0.7, which are presented as bold type in table 4. The increase results from the bigger rock grains' re-breakage under compression, which produces secondary rock grains. Contrarily, though the compression induces re-breakage and produces new secondary rock grains, meanwhile, the compression also promotes the amount of mass loss, but the effect of promoting mass loss is greater than that of re-breakage, which causes the decreases in samples with TPEs of 0.6 and 0.7.

The residual mass of rock grains of 0–2.5 mm in samples with initial compressions is shown in figure 13. It could be discovered that initial compressions have the different effects on samples with different TPEs; more accurately, they have different effects on the re-breakage and mass loss. Taking samples with TPE of 0.1 as an example, due to the high content of the fine rock grains, which are filled in pores and gaps in samples, the initial compressions bring a similar re-breakage effect, but they have different effects on the mass loss. As seen in figure 13, the initial compressions of 10 and 20 mm have similar effects on mass loss, and these samples are relatively loose; thus, much more fine rock grains are lost and the residual masses are less. But samples with the initial compression of

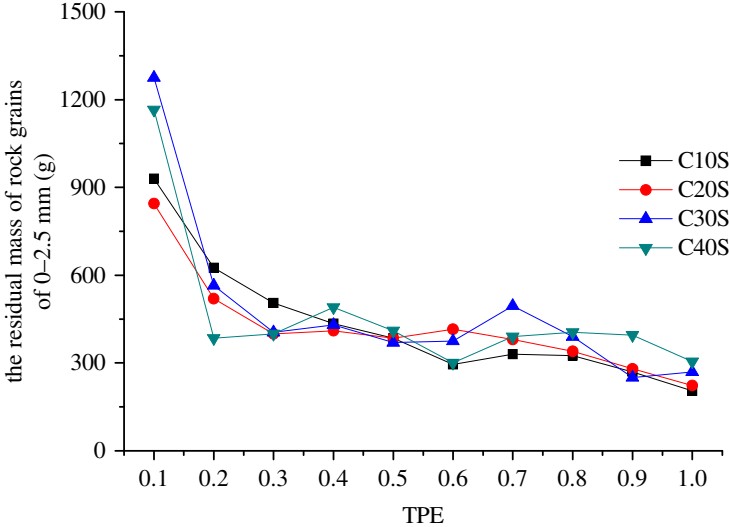

**Figure 13.** The residual mass of rock grains of 0–2.5 mm.

**Table 5.** The mass variation of rock grains with a size of 20–25 mm (g).

| samples | TPE | | | | | | | | | |
|---|---|---|---|---|---|---|---|---|---|---|
| | 0.1 | 0.2 | 0.3 | 0.4 | 0.5 | 0.6 | 0.7 | 0.8 | 0.9 | 1.0 |
| 0S | 44.1 | 87.3 | 129.5 | 170.8 | 211.1 | 250.6 | 289.2 | 327 | 363.9 | 400 |
| C10S | 12 | *0* | 10 | 20 | *0* | 75 | 40 | 115 | 135 | 225 |
| C20S | 43 | 25 | *0* | 45 | 25 | 85 | 10 | 195 | 305 | 300 |
| C30S | 5 | 15 | 60 | 15 | 90 | 60 | 75 | 115 | 210 | 335 |
| C40S | 10 | 15 | 85 | *0* | 130 | 35 | 20 | 75 | 55 | 235 |

30 and 40 mm have worse effects on mass loss, because those samples are relatively dense, leading to less lost mass, and the residual masses are larger. However, in samples with TPE of 1.0, due to the high content of the big rock grains, the initial compressions bring the gradual re-breakage effect and the similar mass loss effect, hence, the residual fine rock grains increase with the initial compressions increasing.

### 3.3.2. The effect of re-breakage of the rock grains

If we want to discuss the effect of re-breakage, we must analyse the residual masses of large rock grains, e.g. 20–25 mm. Under the dual functions of hydration effect and initial compressions, the residual biggest rock grains (20–25 mm) decrease, even to zero, which are presented as italics in table 5.

However, the randomness of rock grains' arrangement in the cylinder has a significant influence on the experimental results [66,67], hence, it can only obtain a few valuable results about the influence of initial compression on re-breakage of the biggest rock grains.

If ignoring the residual mass of C10S, samples (including C20S, C30S and C40S) with TPEs of 0.2, 0.4, 0.6, 0.8 and 0.9 have similar characteristics; the re-breakage continues with the increase of the initial compressions. As initial compression continues, the variation develops towards the ultimate state and rock grains become less likely to be further broken.

For example, the residual mass of C20S with TPE of 0.2 is 25 g, and it decreases to 15 g when in C30S (table 5), but the re-breakage behaviour of the biggest rock grains does not last in C40S, because too many fine rock grains fill in the pores and gaps, and tightly wrap with the bigger rock grains. When it is compressed, the bigger grains move relative to the fine rock grains, and squeeze out from the wrapped fine rock grains. However, due to the difference in the size of the bigger rock grains and the

fine ones, the collisions between them cannot cause the bigger rock grains to re-break, therefore it is impossible for bigger rock grains to break again.

For samples with TPE of 0.4, the re-breakage effect of initial compression has obvious stages, as seen in table 5, the re-breakage effect is better in the process when the initial compression increases from 20 to 30 mm than that from 30 to 40 mm, and we can infer that it has the best re-breakage effect in the process when the initial compression increases from 0 to 10 mm for the samples with TPE of 0.4. Of course, samples with TPE of 0.6 also have such a trend of re-breakage effect.

It seems that the re-breakage behaviour of the samples with TPE of 0.8 varies uniformly during the initial compaction increasing process. Unlike the other samples, samples with TPE of 0.9 have a different re-breakage process when the initial compression is loaded. As seen in table 5, the main re-breakage in samples with TPE of 0.9 occurs in the process when the initial compression increases from 30 to 40 mm.

The analysis above shows that the breakages of rock grains with different sizes have different possibilities, which is inconsistent with Epstein's statement, who once assumed the same probability of breakage for various sizes of grain in 1947 [68]. Generally, a bigger rock grain is easier to break than a smaller one due to the higher probability of defects in bigger grains. However, this is not always true. For example, sometimes bigger rock grains might be surrounded by smaller grains as they migrate into the pore space among the bigger grains, and the smaller grains become a sort of cushion to prevent the further breakage of the bigger grains.

### 3.3.3. The effect of water–rock chemical reactions

Comparing the masses in OS and WCS, the residual mass in most samples reduces except for samples with TPE of 0.9, which is presented as italics in table 4. The reduced fine rock grains result from the mass loss in the seepage process. The increase comes from newly produced fine rock grains in the water–rock reactions, because there is no re-breakage effect in WCS. In a neutral environment, water–rock chemical reactions mainly come from hydration reactions as follows:

$$CaCO_3 + CO_2 + H_2O \rightarrow Ca^{2+} + 2HCO_3^- \tag{3.2}$$

and

$$CaSO_4 \cdot 2H_2O \rightarrow Ca^{2+} + SO_4^{2-} + 2H_2O. \tag{3.3}$$

Of course, if the water is in an acid or alkaline environment, more reactions would happen, for example

$$CaCO_3 + 2H^+ \rightarrow Ca^{2+} + H_2O + CO_2 \uparrow, \tag{3.4}$$

$$CaMg(CO_3)_2 + 4H^+ \rightarrow Ca^{2+} + Mg^{2+} + 2H_2O + 2CO_2 \uparrow, \tag{3.5}$$

$$KAl_3Si_3O_{10}(OH)_2 + 10H^+ \rightarrow 3Al^{3+} + 3H_4SiO_4 + K^+, \tag{3.6}$$

$$SiO_2 + 2OH^- \rightarrow SiO_3^{2-} + H_2O \tag{3.7}$$

and

$$KAl_3Si_3O_{10}(OH)_2 + 8OH^- + H_2O \rightarrow 3Al(OH)_4^- + 3SiO_3^{2-} + K^+. \tag{3.8}$$

In the reactions, water, carbon dioxide, acid or alkali react with minerals in rock grains, forming solutions with many kinds of ions, e.g. $Ca^{2+}$, $HCO_3^-$ $SO_4^{2-}$. The grains are eroded, even broken, and their sizes decrease, also producing many undissolved smaller grains.

In our experiments, the water environment is neutral, the hydration reactions only happen in very small quantities of minerals, and the produced new fine rock grains are just a few. In addition, with the decrease of the content of fine rock grains, the pores and gaps in the samples are more and more; as a result, the areas of the contact surface between water and rock grains gradually increase, so as the hydration effect.

## 4. Discussion

Since the masses of rock grains with different sizes change after the seepage process under the MHC coupling effect, a new expression is needed to describe the varied GSD. Referring to the Talbot continuous grading formula, the ratio $d_i/d_M$ is introduced as a parameter to establish an expression for the varied GSD.

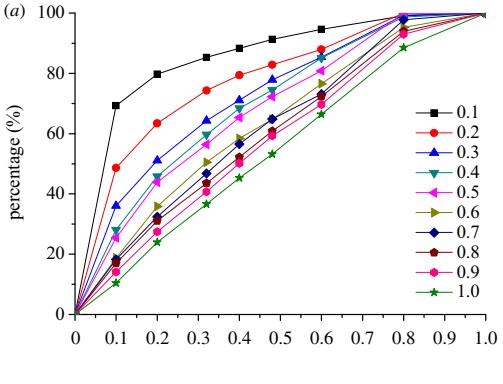

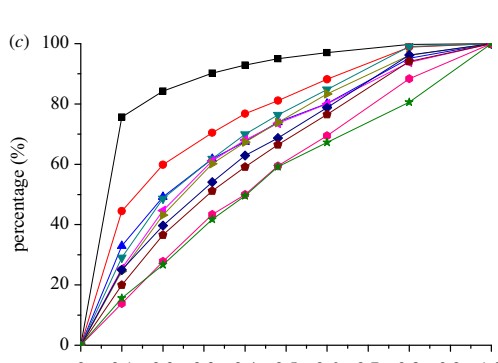

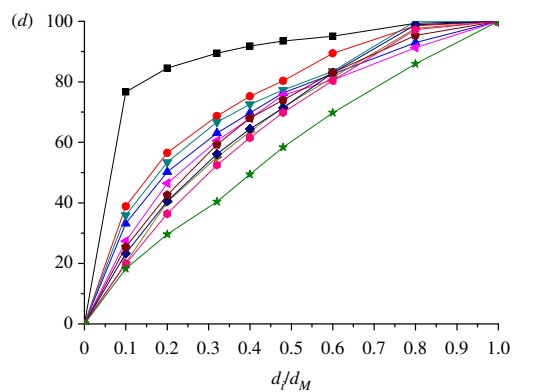

**Figure 14.** The GSD in samples with different TPEs. Samples with initial compression of (*a*) 10 mm, (*b*) 20 mm, (*c*) 30 mm, and (*d*) 40 mm.

### 4.1. The relationship between GSD and the ratio $d_i/d_M$

The relationships between GSD and the ratio $d_i/d_M$ of samples with different TPEs are illustrated in figure 14. It shows that the percentage decreases with the increase of TPE in samples with the same initial compression, and it also decreases with the increase of the initial compression, except $d_i/d_M \leq$ 0.1; namely, when the rock grains' size is ≤2.5 mm, it originally decreases with the initial compression when it increases from 0 to 20 mm, but it conversely increases when the initial compression continuing increases from 20 to 40 mm.

The content of rock grains with the smallest size becomes less and less with TPE increasing, while the content of the biggest rock grains becomes more and more. Because the smallest rock grains in samples have a supplement from the bigger rock grains' re-breakage, the contents of the smallest rock grains are larger in samples with smaller TPEs, and they have supplement even though they lost a lot during the seepage process; meanwhile, they are supplied due to the bigger rock grains' re-breakage, so that the content in samples with smaller TPEs is still larger than those with larger TPEs. The content of the biggest rock grains in samples with larger TPEs is higher; it is still larger than those with smaller TPEs, even though some of the biggest rock grains are broken during the tests.

The GSD is also affected by the initial compression, especially for samples with larger TPEs, such as 0.7–1.0, the GSD of grains whose sizes are less than 15 mm (corresponding to abscissa is 0.8) decreases obviously with the initial compression increasing.

### 4.2. The expression of the varied GSD

According to figure 14, it is easy to find that there is some functional relationship between the varied GSD and the ratio $d_i/d_M$, therefore we fit the relationships referring to the Talbot continuous grading formula, which could be totally expressed as

$$p(d \leq d_i) = a\left(\frac{d_i}{d_M}\right)^b \times 100\%, \tag{4.1}$$

where $a$ and $b$ are the coefficients of the relationship.

**Table 6.** The coefficients of the fitting formula.

| TPE | initial compression (mm) | $a$ | $b$ | $R^2$ |
|---|---|---|---|---|
| 0.1 | 0 | 1.001 | 0.1473 | 0.9867 |
| | 10 | 1.0216 | 0.1618 | 0.9906 |
| | 20 | 1.0177 | 0.1705 | 0.9814 |
| | 30 | 1.0273 | 0.1247 | 0.9744 |
| | 40 | 1.0139 | 0.1166 | 0.9902 |
| 0.2 | 0 | 1.0325 | 0.3448 | 0.9921 |
| | 10 | 1.0455 | 0.3195 | 0.9885 |
| | 20 | 1.0577 | 0.3767 | 0.9892 |
| | 30 | 1.0504 | 0.3602 | 0.9914 |
| | 40 | 1.0762 | 0.4180 | 0.9834 |
| 0.3 | 0 | 1.0699 | 0.4814 | 0.9873 |
| | 10 | 1.0684 | 0.4599 | 0.9918 |
| | 20 | 1.096 | 0.5067 | 0.9839 |
| | 30 | 1.0411 | 0.4830 | 0.9941 |
| | 40 | 1.0504 | 0.4763 | 0.9898 |
| 0.4 | 0 | 1.0623 | 0.6101 | 0.9897 |
| | 10 | 1.104 | 0.5658 | 0.9851 |
| | 20 | 1.0884 | 0.5493 | 0.9816 |
| | 30 | 1.1047 | 0.5429 | 0.9796 |
| | 40 | 1.0691 | 0.4504 | 0.9851 |
| 0.5 | 0 | 1.1596 | 0.7246 | 0.969 |
| | 10 | 1.1017 | 0.6056 | 0.9857 |
| | 20 | 1.0569 | 0.4865 | 0.9935 |
| | 30 | 1.0938 | 0.5866 | 0.9747 |
| | 40 | 1.0732 | 0.5514 | 0.9786 |
| 0.6 | 0 | 1.0358 | 0.6252 | 0.9944 |
| | 10 | 1.1004 | 0.7313 | 0.9885 |
| | 20 | 1.1038 | 0.6096 | 0.9842 |
| | 30 | 1.1185 | 0.6149 | 0.9767 |
| | 40 | 1.1409 | 0.6961 | 0.9755 |
| 0.7 | 0 | 1.1102 | 0.7356 | 0.9913 |
| | 10 | 1.096 | 0.7609 | 0.9931 |
| | 20 | 1.1128 | 0.7038 | 0.9897 |
| | 30 | 1.0721 | 0.6187 | 0.9945 |
| | 40 | 1.1215 | 0.6494 | 0.9849 |
| 0.8 | 0 | 1.0397 | 0.8183 | 0.9981 |
| | 10 | 1.0698 | 0.7861 | 0.9965 |
| | 20 | 1.0607 | 0.7631 | 0.9961 |
| | 30 | 1.0898 | 0.7047 | 0.9905 |
| | 40 | 1.1083 | 0.6038 | 0.9810 |

(*Continued.*)

| TPE | initial compression (mm) | a | b | $R^2$ |
|---|---|---|---|---|
| 0.9 | 0 | 0.99554 | 0.7214 | 0.9981 |
| | 10 | 1.0854 | 0.8680 | 0.9960 |
| | 20 | 1.0213 | 0.8152 | 0.9368 |
| | 30 | 1.0764 | 0.8583 | 0.9940 |
| | 40 | 1.1314 | 0.7193 | 0.9851 |
| 1.0 | 0 | 1.023 | 1.0195 | 0.9980 |
| | 10 | 1.0848 | 0.9818 | 0.9948 |
| | 20 | 1.0337 | 0.9345 | 0.9990 |
| | 30 | 1.0129 | 0.8059 | 0.9967 |
| | 40 | 1.0007 | 0.7508 | 0.9979 |

As seen in table 6, the coefficient $a$ is almost 1.0, and the error results from the uncertainty in the experimental process of crushed rock. Thus, equation (4.1) will be simplified as

$$p(d \leq d_i) = \left(\frac{d_i}{d_M}\right)^b \times 100\%, \tag{4.2}$$

where the coefficient $b$ is related to the lost mass ratio $r_l$ and the initial compression, the lost mass ratio $r_l$ is expressed as

$$r_l = \frac{\sum_{i=1}^{j} m_{li}}{M_T} \times 100\%. \tag{4.3}$$

In equation (4.3), $m_{li}$ is the collected lost mass every other time, which is weighed after being collected and dried, and $j$ is the times of collecting the lost mass.

The relationship between the coefficient $b$ and the lost mass ratio $r_l$ is illustrated in figure 15, and the relationships are fitted and listed in table 7.

Therefore, the coefficient $b$ can be expressed totally as

$$b = C_1 r_l + C_2, \tag{4.4}$$

where $C_1$ and $C_2$ are coefficients of equation (4.4), and both of them are related to the initial compression $h_c$

$$C_1 = 0.0002 h_c - 0.0303 \tag{4.5}$$

and

$$C_2 = -0.0033 h_c + 0.9061. \tag{4.6}$$

Applying equations (4.4)–(4.6) into equation (4.2), the GSD after the seepage process could be expressed by the lost mass ratio and the initial compression as

$$p(d \leq d_i) = \left(\frac{d_i}{d_M}\right)^{(0.0002 h_c - 0.0303) r_l - 0.0033 h_c + 0.9061} \times 100\%. \tag{4.7}$$

In order to verify the result of the GSD expression after experiments, taking the samples with the TPE of 0.4 without initial compression and samples with TPE of 0.5 with initial compression of 10 mm as examples, the GSD after seepage process are illustrated in figure 16, of which the GSD after seepage process is obtained by the experiment and calculated by equation (4.7), respectively. It could be seen from figure 16 that, though the curves of the calculated GSD and the tested GSD almost coincide, the GSD still changes after the seepage process, which embodies the significance of researches on GSD in the rock granular materials and crushed rock with the mass loss after seepage process.

As shown in figure 16, the maximum error occurs at abscissa 0.1 in samples with the TPE of 0.4 without initial compression, namely the rock grains of size less than 2.5 mm, achieving 4.52%. The differences between the calculated values and the tested values are larger in larger rock grains in samples with the

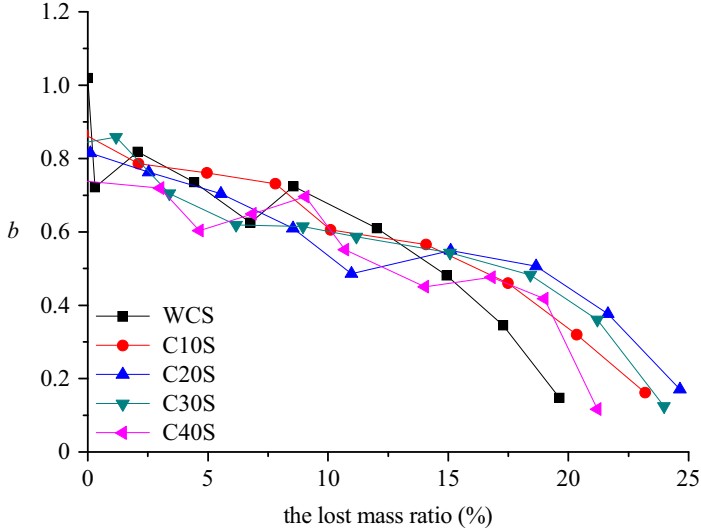

**Figure 15.** The varying characteristic of coefficient $b$.

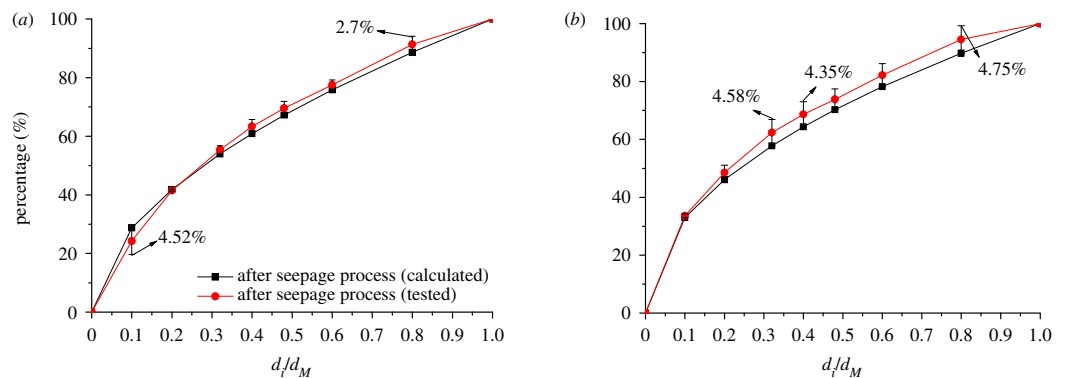

**Figure 16.** Result verification. (*a*) Samples with TPE of 0.4 without initial compression (*b*) Samples with TPE of 0.5 with initial compression of 10 mm.

**Table 7.** The fitting formula of coefficient $b$.

| initial compression (mm) | the formula of the coefficient $b$ | $R^2$ |
|---|---|---|
| 0 | $b = -0.032r_l + 0.8984$ | 0.8565 |
| 10 | $b = -0.0276r_l + 0.8902$ | 0.9640 |
| 20 | $b = -0.0228r_l + 0.8289$ | 0.9210 |
| 30 | $b = -0.0222r_l + 0.8053$ | 0.8753 |
| 40 | $b = -0.0224r_l + 0.7738$ | 0.7810 |

TPE of 0.5 with initial compression of 10 mm. For instance, for the grains of size between 8 and 20 mm, the errors are larger than 4%; the maximum error is 4.75% when the grain size is less than 20 mm. However, the errors are within the permissible range in engineering. It shows that the accuracy of the calculation result is trustworthy based on the result verification and error analysis, and equation (4.7) could be used to calculate the GSD in the rock granular materials under the MHC coupling effect after seepage process.

# 5. Conclusion and expectation

The rock granular material has different sizes, shapes, porosity, arrangement and GSD, which affects its physical and mechanical characteristics. In the nonlinear seepage process, the variation of rock grains'

GSD is very complex, it varies porosity, local stress and seepage fields and may result in seepage catastrophe. It is of great significance and has practical value to study the variation of the GSD of the rock granular materials in the seepage process, and it is very essential to incorporate the varied GSD into the constitutive model in further studies.

In this study, the mass of rock grains after seepage experiments under different compressions is measured and compared with the original ones, the varied GSD is discussed, the MHC coupling effect is analysed, and the expression of the varied GSD is fitted, derived and verified.

(1) The mass of rock grains with different sizes in test samples change during the seepage process, as well as the GSD; they are affected by initial compressions. Comparing with the samples without initial compression, these ones with initial compression of 10 mm have significant changes in the experimental results. The variation of the residual mass ratio has eight types when their initial compressions are different.

(2) Mass loss is the key factor of the mass variation and GSD variation for samples without initial compression. The increase of the residual mass results from the bigger rock grains' re-breakage under compression, which produces secondary rock grains. Compression induces re-breakage and produces new secondary rock grains. Meanwhile, it also promotes the amount of mass loss, and the effect of promoting mass loss is greater than that of re-breakage, which causes the decrease of the residual mass.

(3) The initial compressions have different effects on the re-breakage and mass loss for samples with different TPEs. For samples whose TPEs are lower, they have high contents of the fine rock grains, which are filled in the pores and gaps in samples, the initial compressions bring a similar re-breakage effect. Due to the loose structure, much more fine rock grains are lost and the residual masses are less when the initial compression is 10 and 20 mm. But in samples with the initial compression of 30 and 40 mm, though those samples are relatively dense, they have a worse effect on mass loss. For samples with higher TPEs, due to the high content of big rock grains, the initial compressions bring the gradual re-breakage effect and the similar mass loss effect; hence, the residual fine rock grains increase with the initial compressions increasing.

(4) Samples with TPEs of 0.2, 0.4, 0.6, 0.8 and 0.9 have similar characteristics of re-breakage under initial compressions, the re-breakage continues with the initial compressions increasing. As initial compression continues, the variation develops towards the ultimate state and rock grains become less likely to be further broken. However, the re-breakages of rock grains with different sizes have different possibilities, and their re-breakage stage is related to the compressions.

(5) The increase of the rock grains comes from new produced fine rock grains in the water–rock interaction, because in the samples without initial compression, there is no re-breakage effect. In the reactions, water, carbon dioxide, acid or alkali react with mineral in the rock grains, forming solutions with many kinds of ions and eroding the rock grains, resulting in many undissolved smaller grains.

(6) The mass variation and GSD variation of rock grains both result from the MHC coupling effect, contains the re-breakage induced by mechanical deformation (M), mass loss induced by hydraulic flow (H) and water–rock chemical interaction (C).

(7) The varied GSD of the rock grains after the seepage experiment has a power function relationship with the ratio $d_i/d_M$, and the expression is derived referring to the Talbot continuous grading formula and verified to certify that the accuracy of the calculation result is trustworthy.

The results in this study can be applied to evaluate the change of GSD of the compacted rock granular material in the process of seepage, and to analyse its structural stability and seepage stability under the coupling effect of MHC after more systematic work and wider investigations, and are expected to have a positive impact on further studies of the properties and the recycling of the rock granular materials.

Data accessibility. All revised data are deposited in the Dryad Digital Repository: https://doi.org/10.5061/dryad.nc270f2 [69].

Authors' contributions. H.K. and L.W. carried out the laboratory work, participated in data analysis, participated in the design of the study and drafted the manuscript; H.K. carried out the statistical analyses, L.W. coordinated the study and helped draft the manuscript; H.Z. gave some suggestions and participated in the manuscript revision. All authors gave final approval for publication.

Competing interests. We declare we have no competing interests in this paper.

Funding. This work is supported by the Natural Science Foundation of Jiangsu Province (grant no. BK20160433), the opening fund of Key Laboratory of Safety and High-efficiency Coal Mining, Ministry of Education (Anhui University of Science and Technology) (grant no. JYBSYS2019207), the National Natural Science Fund of China

(grant no. 11502229), the Program of Outstanding Young Scholars in Yancheng Institute of Technology (grant no. 2014), the Program of Yellow Sea Elite in Yancheng Institute of Technology (grant no. 2019) and the Program of Yellow Sea Team in Yancheng Institute of Technology (grant no. 2019).

Acknowledgements. This work was supported by our professional brothers and workmates in China University of Mining and Technology, Anhui University of Science and Technology and Yancheng Institute of Technology, the authors would like to thank the authors of the references, and to acknowledge the editor and three anonymous reviewers for their valuable comments for the improvement of this paper.

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
