## [Reviewer comments · Royal Society Open Science]

Review History

RSOS-190590.R0 (Original submission)

Review form: Reviewer 1

Is the manuscript scientifically sound in its present form?

Yes

Are the interpretations and conclusions justified by the results?

Yes

Is the language acceptable?

Yes

Is it clear how to access all supporting data?

Yes

Do you have any ethical concerns with this paper?

No

Have you any concerns about statistical analyses in this paper?

No

Recommendation?

Accept with minor revision (please list in comments)

Comments to the Author(s)

- 1) It is not clear why "initial compression" is expressed as a length (mm) (page 7, row 25)
- 2) "...water starts to permeate through the specimen for 18000s to record this seepage process as completely as possible..." (page 7, row 37), why the permeate time is 18000s?
- 3) Some of the markers or symbols in the figures are not very clear.
- 4) The statistics of the changes of different initial compression were listed in Table 3. Is there any regularity?
- 5) Errors were presented in Figure 18, and the maximum value is 4.75%. What is the maximum allowable error?

Review form: Reviewer 2

Is the manuscript scientifically sound in its present form?

Yes

Are the interpretations and conclusions justified by the results?

Yes

Is the language acceptable?

Yes

Is it clear how to access all supporting data?

Yes

Do you have any ethical concerns with this paper?

No

Have you any concerns about statistical analyses in this paper?

No

Recommendation?

Accept with minor revision (please list in comments)

Comments to the Author(s)

This paper is interesting and need minor revision.

1. It is not clear enough in legends of Figure 5 and Figure 6, the authors are suggested to modify them.
2. In this paper, the mass variation of rock grains in samples without initial compression and those with initial compression of 10mm were described in Section 3.1.1 and 3.1.2, please explain

why the mass variation of rock grains in samples with the other initial compressions, 20mm, 30mm and 40mm, haven't been described in this paper.

3. The description of the regularity in Figure 14 is not obvious enough.

4. The Section 3.3.3 "The effect of ..." just talks about some "water-rock chemical interaction", could the author quantify these "water-rock chemical interaction"?

5. Have the authors considered that the GSD after text may conform to other distribution rules, not the Talbot continuous grading formula?

6. Table 6 contains too many equations, but they can be explained by Eq. (11). I think the authors can give values of parameters instead of equations.

7. The coefficient a in Eq.(11) then was omitted in Eq. (12), what's the purpose?

Review form: Reviewer 3

Is the manuscript scientifically sound in its present form?

No

Are the interpretations and conclusions justified by the results?

No

Is the language acceptable?

No

Is it clear how to access all supporting data?

Yes

Do you have any ethical concerns with this paper?

No

Have you any concerns about statistical analyses in this paper?

I do not feel qualified to assess the statistics

Recommendation?

Reject

Comments to the Author(s)

The paper presents laboratory investigations on the evolution of grain size distribution in granular systems subjected subsequently to deformation perturbations and then to a water seepage.

In general, it is not recommended to publish this manuscript on Royal Society Open Science. The topic being investigated in this study is interesting, but the technical writing of this manuscript was poor. Technical language and professional terminologies were not properly employed in this manuscript. Multiple grammatical problems and typos were found in the manuscript.

Specific comments are listed below.

1- The introduction may be rewritten. It would be ideal to briefly report the findings/conclusions of the cited studies instead of reporting only their scope.

2- Sec. 2.2: the meaning of "mass distribution of rock grains" is unclear?

3- How was the granular material prepared from the original rock?

- 4- Samples were loaded up to 10, 20, 30, and 40 mm. What is the rate? How did the authors determine these values?
- 5- The specimens are initially loaded by 0.02 MPa. Why? and how was this value determined?
- 6- Sec 3.1: Throughout this section, the authors described their measurements in an overwhelming manner that has rendered difficulties to the reader to follow the content (this is also combined with the poor English used to write the text).
- 7- Sec 4: Despite the various magnitude of "initial" compression, the variation in the GSD is not sharply clear. This is apparent in Table 7, where the expression of b as a function of r is almost similar for all cases. The authors may elaborate on this point.
- 8- Information provided in Table 6 may be presented in a more compact fashion.
- 9- The purpose of the comparison presented in Figure 18 is unclear. Validating models based on the same dataset used to calibrate them does not make any sense.
- 10- The use of "coupling" of MHC along the entire manuscript could be inappropriate.
- 11- Conclusion section needs to be entirely reworded.
- 12- Drawing item 6 in conclusion section from an individual study may be very strong. Establishing a general model may require more systematic work and wider investigations to address aspects that may affect the GSD variation, such as -but not limited to- the shape of the grains, the density of the system, and toughness of the grains surface. In fact, conclusions section needs to be technically improved.
- 13- Information about the aspects mentioned in the previous point was not provided regarding the material used in the study.
- 14- The process of water seepage was not described, and the rate was not provided and justified.
- 15- No need to redefined acronyms (e.g., GSD and MHC) over and over across the manuscript.

Decision letter (RSOS-190590.R0)

13-Aug-2019

Dear Dr Kong,

The editors assigned to your paper ("The variation of grain size distribution (GSD) in the rock granular material in seepage process considering the MHC coupling effect") have now received comments from reviewers. We would like you to revise your paper in accordance with the referee and Associate Editor suggestions which can be found below (not including confidential reports to the Editor). Please note this decision does not guarantee eventual acceptance.

Please submit a copy of your revised paper before 05-Sep-2019. Please note that the revision deadline will expire at 00.00am on this date. If we do not hear from you within this time then it will be assumed that the paper has been withdrawn. In exceptional circumstances, extensions may be possible if agreed with the Editorial Office in advance. We do not allow multiple rounds of revision so we urge you to make every effort to fully address all of the comments at this stage. If deemed necessary by the Editors, your manuscript will be sent back to one or more of the original reviewers for assessment. If the original reviewers are not available, we may invite new reviewers.

- Data accessibility

If you wish to submit your supporting data or code to Dryad (<http://datadryad.org/>), or modify your current submission to dryad, please use the following link:
<http://datadryad.org/submit?journalID=RSOS&manu=RSOS-190590>

- Competing interests

- Authors' contributions

- Acknowledgements

- Funding statement

Kind regards,

Andrew Dunn

on behalf of Professor Ian Guymer (Associate Editor) and R. Kerry Rowe (Subject Editor)

Comments to Author:

Reviewers' Comments to Author:

Reviewer: 1

Comments to the Author(s)

- 1) It is not clear why "initial compression" is expressed as a length (mm) (page 7, row 25)
- 2) "...water starts to permeate through the specimen for 18000s to record this seepage process as completely as possible..." (page 7, row 37), why the permeate time is 18000s?
- 3) Some of the markers or symbols in the figures are not very clear.
- 4) The statistics of the changes of different initial compression were listed in Table 3. Is there any regularity?
- 5) Errors were presented in Figure 18, and the maximum value is 4.75%. What is the maximum allowable error?

Reviewer: 2

Comments to the Author(s)

This paper is interesting and need minor revision.

1. It is not clear enough in legends of Figure 5 and Figure 6, the authors are suggested to modify them.
2. In this paper, the mass variation of rock grains in samples without initial compression and those with initial compression of 10mm were described in Section 3.1.1 and 3.1.2, please explain why the mass variation of rock grains in samples with the other initial compressions, 20mm, 30mm and 40mm, haven't been described in this paper.
3. The description of the regularity in Figure 14 is not obvious enough.
4. The Section 3.3.3 "The effect of ..." just talks about some "water-rock chemical interaction", could the author quantify these "water-rock chemical interaction"?
5. Have the authors considered that the GSD after text may conform to other distribution rules, not the Talbot continuous grading formula?

6. Table 6 contains too many equations, but they can be explained by Eq. (11). I think the authors can give values of parameters instead of equations.
7. The coefficient a in Eq.(11) then was omitted in Eq. (12), what's the purpose?

Reviewer: 3

Comments to the Author(s)

The paper presents laboratory investigations on the evolution of grain size distribution in granular systems subjected subsequently to deformation perturbations and then to a water seepage.

In general, it is not recommended to publish this manuscript on Royal Society Open Science. The topic being investigated in this study is interesting, but the technical writing of this manuscript was poor. Technical language and professional terminologies were not properly employed in this manuscript. Multiple grammatical problems and typos were found in the manuscript.

Specific comments are listed below.

- 1- The introduction may be rewritten. It would be ideal to briefly report the findings/conclusions of the cited studies instead of reporting only their scope.
- 2- Sec. 2.2: the meaning of "mass distribution of rock grains" is unclear?
- 3- How was the granular material prepared from the original rock?
- 4- Samples were loaded up to 10, 20, 30, and 40 mm. What is the rate? How did the authors determine these values?
- 5- The specimens are initially loaded by 0.02 MPa. Why? and how was this value determined?
- 6- Sec 3.1: Throughout this section, the authors described their measurements in an overwhelming manner that has rendered difficulties to the reader to follow the content (this is also combined with the poor English used to write the text).
- 7- Sec 4: Despite the various magnitude of "initial" compression, the variation in the GSD is not sharply clear. This is apparent in Table 7, where the expression of b as a function of r is almost similar for all cases. The authors may elaborate on this point.
- 8- Information provided in Table 6 may be presented in a more compact fashion.
- 9- The purpose of the comparison presented in Figure 18 is unclear. Validating models based on the same dataset used to calibrate them does not make any sense.
- 10- The use of "coupling" of MHC along the entire manuscript could be inappropriate.
- 11- Conclusion section needs to be entirely reworded.
- 12- Drawing item 6 in conclusion section from an individual study may be very strong. Establishing a general model may require more systematic work and wider investigations to address aspects that may affect the GSD variation, such as -but not limited to- the shape of the grains, the density of the system, and toughness of the grains surface. In fact, conclusions section needs to be technically improved.
- 13- Information about the aspects mentioned in the previous point was not provided regarding the material used in the study.
- 14- The process of water seepage was not described, and the rate was not provided and justified.
- 15- No need to redefined acronyms (e.g., GSD and MHC) over and over across the manuscript.

Author's Response to Decision Letter for (RSOS-190590.R0)

See Appendices A - C.

RSOS-190590.R1 (Revision)

Review form: Reviewer 3

Is the manuscript scientifically sound in its present form?

Yes

Are the interpretations and conclusions justified by the results?

Yes

Is the language acceptable?

No

Do you have any ethical concerns with this paper?

No

Have you any concerns about statistical analyses in this paper?

No

Recommendation?

Accept with minor revision (please list in comments)

Comments to the Author(s)

1- Items 3,4 and 5 should be commented in the main text of the manuscript, rather than only in the reply-to-reviewer document.

2- A brief description on water seepage (item 14) should be provided in the main text of the manuscript.

3- The reviewer believes that section 3.1 should be rewritten in a more condensed manner

4- English can be improved by proofreading

Decision letter (RSOS-190590.R1)

28-Oct-2019

Dear Dr Kong:

On behalf of the Editors, I am pleased to inform you that your Manuscript RSOS-190590.R1 entitled "The variation of grain size distribution (GSD) in the rock granular material in seepage process considering the MHC coupling effect" has been accepted for publication in Royal Society Open Science subject to minor revision in accordance with the referee suggestions. Please find the referees' comments at the end of this email.

The reviewers and Subject Editor have recommended publication, but also suggest some minor revisions to your manuscript. Therefore, I invite you to respond to the comments and revise your manuscript.

- Ethics statement

- Data accessibility

<http://datadryad.org/submit?journalID=RSOS&manu=RSOS-190590.R1>

- Competing interests

- Authors' contributions

- Acknowledgements

- Funding statement

Please note that we cannot publish your manuscript without these end statements included. We have included a screenshot example of the end statements for reference. If you feel that a given

heading is not relevant to your paper, please nevertheless include the heading and explicitly state that it is not relevant to your work.

Because the schedule for publication is very tight, it is a condition of publication that you submit the revised version of your manuscript before 06-Nov-2019. Please note that the revision deadline will expire at 00.00am on this date. If you do not think you will be able to meet this date please let me know immediately.

Kind regards,
Anita Kristiansen
Editorial Coordinator
Royal Society Open Science
openscience@royalsociety.org

on behalf of Professor Ian Guymer (Associate Editor) and R. Kerry Rowe (Subject Editor)
openscience@royalsociety.org

Reviewer comments to Author:

Reviewer: 3

Comments to the Author(s)

1- Items 3,4 and 5 should be commented in the main text of the manuscript, rather than only in the reply-to-reviewer document.

2- A brief description on water seepage (item 14) should be provided in the main text of the manuscript.

3- The reviewer believes that section 3.1 should be rewritten in a more condensed manner

4- English can be improved by proofreading

Author's Response to Decision Letter for (RSOS-190590.R1)

See Appendices D & E.

Decision letter (RSOS-190590.R2)

28-Nov-2019

Dear Dr Kong,

It is a pleasure to accept your manuscript entitled "The variation of grain size distribution in rock granular material in seepage process considering the mechanical-hydrological-chemical coupling effect" in its current form for publication in Royal Society Open Science. The comments of the reviewer(s) who reviewed your manuscript are included at the foot of this letter.

on behalf of Professor Ian Guymer (Associate Editor) and R. Kerry Rowe (Subject Editor)
openscience@royalsociety.org

Follow Royal Society Publishing on WeChat
Follow Royal Society Publishing on Twitter: @RSocPublishing
Follow Royal Society Publishing on Facebook:
<https://www.facebook.com/RoyalSocietyPublishing.FanPage/>
Read Royal Society Publishing's blog: <https://blogs.royalsociety.org/publishing/>

Appendix A

Reply to the editor

Dear editor

Have a good day.

We all very glad to receive the decision letter from the Royal Society Open Science Editorial Office, and thank you for giving us a chance to revise the manuscript.

Based on the reviewers' comments, we revised the manuscript and resubmitted a revision, in which the modified parts were highlighted in yellow, and the deleted parts were marked with deleted line in blue.

The authors replied to the reviewers' comments one by one in the attachment "Reply to the reviewers' comments". Some important revisions are explained as follow.

(1) Hualei Zhang, who gave suggestions on this article and participated in manuscript revision, therefore, we add him as the third co-author as well the co-corresponding author in the revised manuscript.

(2) The authors added two funds from the author Hailing Kong, *the open funding of Key Laboratory of Safety and High-efficiency Coal Mining, Ministry of Education of China and the Program of Yellow Sea Team in Yancheng Institute of Technology.*

(3) The authors deleted Figure 5, Figure 14 in the original manuscript, and modified the expression form of Table 6.

PS: there's still a little problem in the reference that the authors' work "*Kong HL, Wang LZ. 2019 The behavior of mass migration and loss in fractured rock during seepage. Bulletin of Engineering Geology and the Environment. 78, 16pages.*" has published online on 18 July 2019, but it hasn't got its issue and page.

We all hope the revised manuscript can meet the requirements of the Royal Society Open Science.

Best wishes.

Hailing Kong, Luzhen Wang, Hualei Zhang

Appendix B

Authors Adjustment Note

The authors would like to add Hualei Zhang to the authors list in this manuscript, "The variation of grain size distribution (GSD) in the rock granular material in seepage process considering the MHC coupling effect: An experimental research". Because from the very beginning, Hualei Zhang has participated in this study, given us a lot of advice on this article and worked on the paper during the review process. With the approval of all the other co-authors, we would like to add Hualei Zhang as a new co-author in this manuscript.

Considering that the manuscript is still in editing process, we hope this would be okay for you and we would really appreciate it.

Thanks again!

Harling Kong
Luzhen Wang
Hualei Zhang

Appendix C

Reply to the reviewers' comments

Reviewer: 1

1) It is not clear why “initial compression” is expressed as a length (mm) (page 7, row 25)

Reply: The authors found that the reviewer's doubts might be due to the authors' lack of clarity in the description of initial compression.

To analyze the influence of initial compression on mass migration and loss, samples were compressed to a certain degree by axial loading. In our tests, the axial loading displacements were 0, 10, 20, 30, and 40 mm. So that the compression was achieved in this experiment by applying a certain amount of axial displacement. The different “length (mm)” was actually different initial compression conditions.

The authors hoped that such a description would no longer be ambiguous and would no longer be comprehensible to the reader.

2) “...water starts to permeate through the specimen for 18000s to record this seepage process as completely as possible...” (page 7, row 37), why the permeate time is 18000s?

Reply: In order to simulate the water seepage process in fractured rock in actual engineering, which will last for days, months or even years, the seepage process in our experiment should last for a longer time. The long-term in this paper is really a relative definition, the permeate time measured in this paper and reaches 18000s. Actually, the seepage system could realize a longer time.

3) Some of the markers or symbols in the figures are not very clear.

Reply: This phenomenon does exist, probably when WORD is transferred to PDF. The authors will check it before resubmit the revised manuscript.

4) The statistics of the changes of different initial compression were listed in Table 3. Is there any regularity?

Reply: The authors studied the changes of different initial compression from the viewpoint of statistical analysis of experimental results, and we also hoped that the statistical analysis result would have some regularity, but we can only explain the possibility of various changes from Table 3, and no more regularity has been found, which is regrettable.

5) Errors were presented in Figure 18, and the maximum value is 4.75%. What is the maximum allowable error?

Reply: At present, no clear and existing standard has established to stipulate the allowable range of error value in this research content, so we can apply the allowable range of quality error in general industry, which is $\pm 5\%$. Obviously, the error between the calculated value and test value is 4.75% which meets the allowable range of $\pm 5\%$.

Reviewer: 2

This paper is interesting and need minor revision.

1. It is not clear enough in legends of Figure 5 and Figure 6, the authors are suggested to modify them.

Reply: This phenomenon does exist, probably when WORD is transferred to PDF. The authors will check it before resubmitting the revised manuscript.

2. In this paper, the mass variation of rock grains in samples without initial compression and those with initial compression of 10mm were described in Section 3.1.1 and 3.1.2, please explain why the mass variation of

rock grains in samples with the other initial compressions, 20mm, 30mm and 40mm, haven't been described in this paper.

Reply: In order to study the difference between the influences on the mass variation of rock grains of the initial compression, the authors compared the mass variation of rock grains in samples without initial compression and those with initial compression, in this paper the authors chose the result of the samples with initial compression of 10mm to show the readers the influences on the mass variation of rock grains of the initial compression. Of course, the authors also could choose samples with the other initial compressions, 20mm, 30mm, and 40mm. Due to the limitation of the length of the article, it was impossible to give a detailed description of the variation under various compressions, so only samples with initial compression of 10mm were selected to describe in this manuscript.

3. The description of the regularity in Figure 14 is not obvious enough.

Reply: It seems that the explanation is not obvious enough if the readers look at Figure 14 alone. Actually, the authors studied the changes of different initial compression from the viewpoint of statistical analysis of experimental results combining Figure 14 and Table 3. We hoped that the description of Figure 14 and Table 3 could afford enough obvious information for the readers.

4. The Section 3.3.3 "The effect of ..." just talks about some "water-rock chemical interaction", could the author quantify these "water-rock chemical interaction"?

Reply: The authors thought that the mass variation resulted from the MHC coupling effect, including "the effect of water-rock chemical interaction". However, the MHC coupling effect was a kind of combination, it's difficult to separate the three kinds of effect lonely strictly. Therefore, in this manuscript, the authors just described the effect of water-rock chemical interaction qualitatively, but not quantitatively.

5. Have the authors considered that the GSD after test may conform to other distribution rules, not the Talbot continuous grading formula?

Reply: The authors don't agree with the reviewer's opinion that GSD after test may conform to other distribution rules, not the Talbot continuous grading formula. In this manuscript, the authors investigated the test data, and found that the GSD after test also conformed to the Talbot continuous grading formula, please see Eq. (11), (12) and Table 6.

6. Table 6 contains too many equations, but they can be explained by Eq. (11). I think the authors can give values of parameters instead of equations.

Reply: Thanks for the reviewer's suggestion, it indeed contains too many equations, and the authors have given values of parameters of Eq. (11) instead of listing all equations.

7. The coefficient a in Eq.(11) then was omitted in Eq. (12), what's the purpose?

Reply: As seen in Table 6, the value of the coefficient a is always between 0.99554 and 1.1596, which are around 1, the error may result from the uncertainty in the experiment process of fractured rock. Therefore, the authors considered to omit it in Eq. (12).

Reviewer: 3

1- The introduction may be rewritten. It would be ideal to briefly report the findings/conclusions of the cited studies instead of reporting only their scope.

Reply: Thanks for the reviewer's suggestion, the authors have rewritten the introduction, and added the findings of the cited studies.

2- Sec. 2.2: the meaning of "mass distribution of rock grains" is unclear?

Reply: It is not clear why reviewers still have doubts about the mass distribution of this manuscript? The mass of various particles in various proportions was clearly listed in the table.

3- How was the granular material prepared from the original rock?

Reply: Mudstone samples used in this test were collected from Changcun Coal Mine, Lu'an Mining Bureau, Shanxi Province, China. The rock grain materials were screened from mudstone after crushing.

4- Samples were loaded up to 10, 20, 30, and 40 mm. What is the rate? How did the authors determine these values?

Reply: The loading rate in our experiment was 1mm/min. In order to analyze the influence of initial compression on mass migration and loss, samples were compressed to a certain degree by axial loading. In our tests, the axial loading displacement was 0, 10, 20, 30, and 40 mm. Because the mass of rock grains will have obvious variation when the loading gradient is set every 10mm through the pre-test, and the authors also found that the loading gradient of 5mm could not achieve such a significant effect, and the discrimination was not significant enough. If the loading gradient was set as every 15mm or 20mm, it was difficult to achieve 3-4 different compaction quantities in the test, and it was difficult to compare, too. So, the loading gradient of 10 mm is relatively reasonable.

5- The specimens are initially loaded by 0.02 MPa. Why? and how was this value determined?

Reply: In order to simulate the state of fractured rock mass in engineering, the samples would be pre-loaded certain pressure, on the one hand, this pressure cannot make the structure of fractured rock mass deform, on the other hand, it cannot make the fractured rock particles compress and destroy. Through many experiments, the authors found that the pressure applied can meet both of the above requirements when it does not exceed 0.02MPa.

6- Sec 3.1: Throughout this section, the authors described their measurements in an overwhelming manner that has rendered difficulties to the reader to follow the content (this is also combined with the poor English used to write the text).

Reply: Section 3.1 is the very important part of this manuscript, therefore, the authors used a lot of words to describe it. We are sorry to bring difficulties to the readers to follow the content, but it's necessary to use such lengthy content to describe it clearly.

The authors have to say sorry to show the reviewers such a poor English manuscript. The authors are not from a native English-speaking country. In the process of writing this manuscript, the expression is inevitably influenced by the authors' native language and thinking, so that some parts in the original manuscript are not easily readable. The authors have improved the English writing in the revised manuscript. And the authors hoped that the improved manuscript could meet the language requirements of the reviewers, readers and this Journal.

7- Sec 4: Despite the various magnitude of “initial” compression, the variation in the GSD is not sharply clear. This is apparent in Table 7, where the expression of b as a function of r is almost similar for all cases. The authors may elaborate on this point.

Reply: Actually, the effect of initial compression on GSD is still obvious, especially when the match ratio (TPE) is large, the GSD of samples with different TPEs under different compressions is given in the figure below, which is afforded to the reviewer as a reference. The figure is similar to Figure 16 in the manuscript, therefore, the authors will not add the figure in the article. The expressions of the coefficient b in Table 7 is almost similar for all cases, which just shows the readers that it has some regularity, doesn't it?

(a) 0.1

(b) 0.2

(c) 0.3

(d) 0.4

(e) 0.5

(f) 0.6

(g) 0.7

(h) 0.8

(i) 0.9

(j) 1.0

Figure The GSD in samples with different compressions

8- Information provided in Table 6 may be presented in a more contract fashion.

Reply: Thanks for the reviewer's suggestion, it indeed contains too many equations, and the authors have given values of parameters of Eq. (11) instead of listing all equations.

9- The purpose of the comparison presented in Figure 18 is unclear. Validating models based on the same dataset used to calibrate them does not make any sense.

Reply: The purpose of Fig. 18 is to compare and verify the expression of grain size distribution after the test established by the author. It is incomprehensible why the reviewer considered it meaningless.

10- The use of "coupling" of MHC along the entire manuscript could be inappropriate.

Reply: The authors thought that the mass variation resulted from the MHC coupling effect, which was a continuous process coupled by mechanical effect, hydrological effect and chemical effect. Therefore, the authors considered that coupling was more suitable than coupled in this manuscript.

11- Conclusion section needs to be entirely reworded.

Reply: Thanks for the reviewer's suggestion, the authors have rewritten the conclusion.

12- Drawing item 6 in conclusion section from an individual study may be very strong. Establishing a general model may require more systematic work and wider investigations to address aspects that may

affect the GSD variation, such as -but not limited to- the shape of the grains, the density of the system, and toughness of the grains surface. In fact, conclusions section needs to be technically improved.

Reply: As the reviewer said, “establishing a general model may require more systematic work and wider investigations to address aspects that may affect the GSD variation, such as -but not limited to- the shape of the grains, the density of the system, and toughness of the grains surface”, the authors did not limit the influence factors to the shape of the grains, the density of the system, etc., but due to the manuscript, these factors were introduced to show their effect on the GSD variation. Furthermore, it's unrealistic to consider all the factors.

The authors have rewritten the item 6 in the conclusion.

13- Information about the aspects mentioned in the previous point was not provided regarding the material used in the study.

Reply: The information about the test and materials have been described in Section 2.

14- The process of water seepage was not described, and the rate was not provided and justified.

Reply: On one hand, in this manuscript, the process of water seepage was not described, because it wasn't the subject of this manuscript. On the other hand, the detail seepage experiment process has been described in another paper which was the reference [49] (H.L. Kong, L.Z. Wang, The mass loss behavior of fractured rock in seepage process: The development and application of a new seepage experimental system, *Advances in Civil Engineering* 2018(7891914) (2018) 1-12.) and had been published in 2018.

15- No need to redefined acronyms (e.g., GSD and MHC) over and over across the manuscript.

Reply: The reviewer thought that the acronyms (e.g., GSD and MHC) were not needed over and over across the manuscript, but the authors had a decidedly different opinion about the using of these acronyms (e.g., GSD and MHC). Firstly, these acronyms (e.g., GSD and MHC) are not originated by the authors, they can be seen in many other literatures. Secondly, using these acronyms (e.g., GSD and MHC) will provide readers with a lot of conveniences. Thirdly, this is the embodiment of specialization.

Appendix D

Reply to the editor

Dear editor

Have a good day.

We are all very glad to receive the decision letter from the Royal Society Open Science Editorial Office, and thank you for accepting the manuscript with minor revision.

Based on the reviewers' comments, we revised the manuscript and resubmitted a revision, in which the modified parts were highlighted in yellow.

The authors replied to the reviewers' comments one by one in the attachment "Reply to the reviewers' comments-v2".

1- Items 3,4 and 5 should be commented in the main text of the manuscript, rather than only in the reply-to-reviewer document.

Reply: Thanks for the reviewer's suggestion, the items 3, 4 and 5, which are talked about the granular materials' preparation, the initial compressions and the initial loading pressure, have been commented in the main text of the revised manuscript. These parts have been marked in yellow in the revision.

2- A brief description on water seepage (item 14) should be provided in the main text of the manuscript.

Reply: Thanks for the reviewer's suggestion. In order to show the process of water seepage to the readers, the authors added a reference (*H.L. Kong, L.Z. Wang, The mass loss behavior of fractured rock in seepage process: The development and application of a new seepage experimental system, Advances in Civil Engineering 2018(7891914) (2018) 1-12.*) about it, which can show the readers the details of the seepage experiment process

3- The reviewer believes that section 3.1 should be rewritten in a more condensed manner

Reply: Thanks for the reviewer's suggestion. The authors have read the full text carefully again and discussed the reviewer's opinion to rewrite section 3.1 in a more condensed manner. Finally, the authors decided not to abridge the section, because this section is closely related to the following analysis.

Of course, the reviewer's opinion is worth considering partly that the content of section 3.1 is quite long and bring difficulties to the readers, so the authors adjusted the structure of the article slightly.

Section 3.1 has been divided into two parts in the revised manuscript, sections 3.1 and 3.2, which talked about the mass variation before and after seepage experiments and the variation of GSD in samples with different initial compressions.

After the adjustment, the structure of the article is more reasonable, and the readers are more likely to understand the information that the article wants to convey to them.

4- English can be improved by proofreading

Reply: Thanks for the reviewer's tolerance and understanding of the English language expression in this manuscript. The authors have proofread the whole manuscript and improved English language expression.

We all hope the revised manuscript can meet the requirements of the Royal Society Open Science.

Best wishes.

Hailing Kong, Luzhen Wang, Hualei Zhang

Appendix E

Reply to the reviewers' comments

The authors replied to the reviewers' comments one by one as follow.

1- Items 3,4 and 5 should be commented in the main text of the manuscript, rather than only in the reply-to-reviewer document.

Reply: Thanks for the reviewer's suggestion, the items 3, 4 and 5, which are talked about the granular materials' preparation, the initial compressions and the initial loading pressure, have been commented in the main text of the revised manuscript. These parts have been marked in yellow in the revision.

2- A brief description on water seepage (item 14) should be provided in the main text of the manuscript.

Reply: Thanks for the reviewer's suggestion. In order to show the process of water seepage to the readers, the authors added a reference (*H.L. Kong, L.Z. Wang, The mass loss behavior of fractured rock in seepage process: The development and application of a new seepage experimental system, Advances in Civil Engineering 2018(7891914) (2018) 1-12.*) about it, which can show the readers the details of the seepage experiment process

3- The reviewer believes that section 3.1 should be rewritten in a more condensed manner

Reply: Thanks for the reviewer's suggestion. The authors have read the full text carefully again and discussed the reviewer's opinion to rewrite section 3.1 in a more condensed manner. Finally, the authors decided not to abridge the section, because this section is closely related to the following analysis.

Of course, the reviewer's opinion is worth considering partly that the content of section 3.1 is quite long and bring difficulties to the readers, so the authors adjusted the structure of the article slightly.

Section 3.1 has been divided into two parts in the revised manuscript, sections 3.1 and 3.2, which talked about the mass variation before and after seepage experiments and the variation of GSD in samples with different initial compressions.

After the adjustment, the structure of the article is more reasonable, and the readers are more likely to understand the information that the article wants to convey to them.

4- English can be improved by proofreading

Reply: Thanks for the reviewer's tolerance and understanding of the English language expression in this manuscript. The authors have proofread the whole manuscript and improved English language expression.